# Partial Identification under High-Dimensional Potential Outcomes and Confounders via Optimal Transport

Yunfeng Wang [1]   Zhiheng Zhang[✉ 2 3]   Zijun Gao [4]

## Abstract

Partial identification provides informative causal guarantees when point identification is impossible, but existing approaches based on optimal transport (OT) become computationally and statistically intractable in high-dimensional settings. This limitation is particularly severe when both potential outcomes and confounders are high-dimensional, where classical OT-based bounds suffer from the curse of dimensionality and unfavorable convergence rates. To address this challenge, we propose a novel estimator that decomposes the transport problem into a low-dimensional signal subspace and a high-dimensional residual subspace. Unlike existing projection-based methods that discard residual information, we recover the residual transport energy using the Sliced Wasserstein distance, which is computationally efficient and robust to high dimensions. We establish interpretable conditions controlling the approximation gap based on residual structure and provide a data-driven rule for signal dimension selection. Empirical results show that our estimator consistently outperforms projection-only baselines by recovering lost transport energy, yielding more informative causal bounds while remaining computationally tractable in high dimensions.

## 1 Introduction

Causal inference aims to quantify the effect of interventions from incomplete data. In many settings, point identification is unattainable: causal estimands depend on the joint law of potential outcomes, while the observed data only reveal marginal or conditionally marginal distributions. This places a broad class of problems in the domain of *partial identification* (PI), where the goal is to characterize the set of causal effects consistent with observed information and minimal structural assumptions. Partial identification plays a central role in modern causal analysis, providing robustness against untestable assumptions and enabling principled decision-making under ambiguity.

Pre-treatment covariates can substantially tighten PI sets by restricting the feasible dependence between counterfactual variables. Recently, a series of works has established a deep connection between covariate-assisted PI and optimal transport (OT). Model-agnostic inference frameworks based on OT duality deliver valid causal bounds even when conditional distributions are estimated imprecisely, and admit doubly robust extensions in observational studies (Ji et al., 2024). Related work shows that covariate-assisted PI can be characterized through conditional optimal transport (COT), and that tight relaxations reduce COT to standard OT with explicit penalties, yielding increasingly sharp bounds (Lin et al., 2025b). A fundamental difficulty—discontinuity of COT under weak convergence—has been resolved by establishing continuity under adapted Wasserstein topologies, leading to consistent nonparametric estimation of optimal causal bounds (Lin et al., 2025a). Together, these results identify conditional transport discrepancies as the correct geometric objects for covariate-assisted PI.

Despite this progress, a critical challenge remains. In modern applications, covariates and learned representations are often high-dimensional, while conditional OT functionals are notoriously difficult to estimate reliably in such regimes. Conditioning further reduces effective sample sizes, and direct plug-in estimation of conditional Wasserstein or COT quantities can be statistically unstable or computationally prohibitive. As a result, existing covariate-assisted OT methods face a common failure mode in practice: either the bounds become overly conservative due to coarse stratification, or they rely on modeling assumptions that undermine robustness. This reveals a key research gap: *how to construct dimension-adaptive, statistically stable transport quantities that can be used within covariate-assisted PI*

[1]Fudan University [2]School of Statistics and Data Science, Shanghai University of Finance and Economics, Shanghai, China [3]Institute of Big Data Research, Shanghai University of Finance and Economics, Shanghai, China [4]Department of Data Science and Operations, University of Southern California, USA. Correspondence to: Zhiheng Zhang <zhangzhiheng@mail.shufe.edu.cn>.

*Proceedings of the $43^{rd}$ International Conference on Machine Learning*, Seoul, South Korea. PMLR 306, 2026. Copyright 2026 by the author(s).

*frameworks without sacrificing theoretical validity.*

We address this gap by developing a conditioned, dimension-adaptive lower bound for quadratic Wasserstein distances in high-dimensional settings. Our approach is motivated by the observation that, in many causal problems, distributional differences between treatment and control are *anisotropic* and concentrated on a low-dimensional subspace, with the remaining high-dimensional variation behaving approximately isotropically. Building on this structure, we propose a *conditioned subspace–slicing* (CSS) decomposition. It combines a projected Wasserstein term on a learned low-dimensional signal subspace with a scaled sliced-Wasserstein correction on the residual space. This estimates the difficult transport component in low dimension while recovering residual energy through one-dimensional projections.

Our first main result shows that this quantity yields a valid conditional lower bound. To formalize when the bound is informative, we introduce an *extended spiked transport model* (ESTM), which generalizes spiked covariance models to optimal transport. Under ESTM, transport energy concentrates on a low-dimensional signal subspace, and we show that Wasserstein projection pursuit (WPP) recovers this subspace as the optimizer of the CSS bound. The remaining gap between the true Wasserstein distance and the optimized lower bound is governed by a single residual anisotropy term. A key and somewhat counterintuitive insight is that, once appropriately scaled, random one-dimensional projections can recover the full residual transport energy when the residual component is close to isotropic. This yields a principled "free lunch": replacing a high-dimensional transport problem with a combination of low-dimensional OT and inexpensive sliced computations, without loosening the bound under mild conditions.

We complement the population theory with a finite-sample analysis. We propose a practical estimator based on WPP for the signal term and Monte Carlo sliced Wasserstein estimation for the residual, and derive nonasymptotic error bounds that depend on intrinsic transport dimension rather than ambient dimension. Our contribution is complementary to existing covariate-assisted PI frameworks: while prior work focuses on inferential robustness, relaxation tightness, and topological consistency, we provide a dimension-adaptive geometric primitive that can be plugged into these pipelines when high dimensionality makes direct conditional transport estimation fragile. Our main contributions are summarized as follows:

- We introduce a conditioned subspace–slicing lower bound for quadratic Wasserstein distances, serving as the partial identification region, and yielding certified conditional and unconditional transport lower bounds in high dimensions.

- We formalize an extended spiked transport model under which the bound is near-tight and the optimal subspace is identified by Wasserstein projection pursuit.

- We develop finite-sample estimators with nonasymptotic error guarantees that scale with intrinsic rather than ambient dimension.

**Organization.** Section 2 reviews Wasserstein distance, projection pursuit, and sliced Wasserstein distances. Section 3 develops the conditioned subspace–slicing bound and its structural properties, and further presents finite-sample estimators together with nonasymptotic error bounds. Section 4 presents the experiments on synthetic data. Section A collects proofs.

## 1.1 Literature Review and Positioning

**Partial identification and covariate assistance.** Partial identification provides a principled response to the fact that many causal estimands depend on counterfactual joint laws that are not determined by the observed distribution. Classical work characterizes identified sets under minimal restrictions and highlights the role of auxiliary information in tightening bounds; see, e.g., Manski (2003) and Imbens & Manski (2004). In contemporary applications, covariate information is often rich and continuous, so "conditioning to tighten bounds" becomes an infinite-dimensional operation that must be handled with care: naïve stratification is conservative, while fully parametric modeling may sacrifice robustness (Zhang, 2026; 2024; Zhang & Su, 2024; Zhang et al., 2023).

**OT-based formulations of covariate-assisted causal bounds.** A recent line of work has established that many covariate-assisted PI problems admit a natural representation in terms of optimal transport (OT) and its conditional variants. Ji et al. (2024) develop a model-agnostic approach to inference for covariate-assisted partially identified effects, leveraging OT duality to obtain valid procedures that remain meaningful even when conditional nuisance components are difficult to estimate. Complementarily, Lin et al. (2025b) show that covariate-assisted sharp bounds can be characterized through conditional optimal transport (COT), and propose covariate-aware transport relaxations that reduce COT to standard OT with an explicit penalty, yielding systematically tighter causal bounds as the penalty increases. A further difficulty is that conditional transport functionals can behave pathologically under weak convergence; Lin et al. (2025a) address this by identifying topologies under which the relevant conditional transport quantities become continuous, thereby enabling consistent nonparametric estimation of optimal causal bounds. Taken together, these papers establish OT/COT as the correct geometric primitives for covariate-assisted PI and clarify how sharpness, compu-

tation, and inference interact in this paradigm. Our paper instead tackles the curse of dimensionality through a signal–residual subspace decomposition with a sliced-Wasserstein residual correction.

**High-dimensional transport as the statistical bottleneck.** The above developments also expose a bottleneck that is largely orthogonal to sharpness and inferential robustness: empirical transport in high ambient dimension is statistically fragile. It is well known that plug-in estimators of Wasserstein distances deteriorate rapidly with dimension without additional structure (Fournier & Guillin, 2015; Weed & Bach, 2019), motivating scalable surrogates and dimension-reduction principles. Two prominent paradigms are: (i) *projection pursuit*, which searches for low-dimensional projections that preserve the dominant transport discrepancy (Paty & Cuturi, 2019; Lin et al., 2020), and (ii) *slicing*, which averages one-dimensional transport costs over random directions to obtain computationally tractable quantities (Rabin et al., 2011; Bonneel et al., 2015). A separate, structure-driven approach is the *spiked transport model* of Niles-Weed & Rigollet (2022), which shows that when transport energy concentrates on a low-dimensional subspace, Wasserstein distances can be estimated at intrinsic-dimensional rates.

In OT-based PI pipelines, one typically encounters optimization problems whose objective depends on transport discrepancies between (conditional) laws. In high-dimensional regimes, a natural reduction is to keep only the *dominant low-dimensional signal* by optimizing the projection term $V \mapsto W_2^2(\mu^V, \nu^V)$ over $r$-dimensional subspaces, which aligns with projection-pursuit transport objectives and admits favorable estimation rates. However, this projection-only strategy has an intrinsic limitation that is not merely computational: it yields a *systematically loose lower certificate* whenever there remains non-negligible discrepancy on the orthogonal complement $V^\perp$. In particular, even under benign residual geometries—e.g., approximately isotropic residual transport—the projection-only objective discards a component of the transport energy that can, in principle, be certified from one-dimensional projections.

The present paper adopts a different paradigm from both (i) direct estimation of conditional OT/COT functionals and (ii) projection-only dimension reduction. We couple the two complementary projection principles in OT by considering, for each candidate subspace $V$,

$$W_2^2(\mu^V, \nu^V) \, + \, k \, \mathrm{SW}_2^2(\mu^{V^\perp}, \nu^{V^\perp}), \qquad k = d - r,$$

thereby retaining the low-dimensional transport signal while simultaneously certifying residual discrepancy through a scaled sliced term. This augmentation is not an ad hoc variance-reduction device: $k \, \mathrm{SW}_2^2$ is itself a deterministic lower bound on $W_2^2$ on the residual space, and becomes near-tight under residual isotropy. Consequently, our two-term

objective strictly improves upon projection-only certificates whenever residual discrepancy is present, while preserving statistical scalability because the sliced component is built from one-dimensional OT computations.

**Relation to existing causal-OT frameworks.** Our results are complementary to the covariate-assisted PI literature cited above. Whereas Ji et al. (2024) emphasize inferential validity under minimal modeling and Lin et al. (2025b;a) emphasize sharpness/computation and the topological regularity needed for consistent estimation of conditional transport quantities, we focus on the *dimension-adaptive transport-analytic component* required by these pipelines. Specifically, we provide (i) a population-level upper control of the certificate gap (a "bias" bound relative to the true quadratic transport radius), (ii) finite-sample error bounds for the plug-in implementation, and (iii) a structural theorem identifying the population-optimal subspace under an extended spiked transport model. These components are designed to be modular: they can be inserted as conservative, dimension-adaptive primitives wherever conditional quadratic transport radii enter downstream partially identified causal optimization problems.

## 2 Preliminaries

For $d \geq 1$, let $\langle \cdot, \cdot \rangle$ and $\| \cdot \|$ denote the Euclidean inner product and norm on $\mathbb{R}^d$. Let $\mathcal{P}_2(\mathbb{R}^d)$ be the set of Borel probability measures on $\mathbb{R}^d$ with finite second moment. For a measurable map $T : \mathbb{R}^d \to \mathbb{R}^m$ and $\mu \in \mathcal{P}_2(\mathbb{R}^d)$, we write $T_{\#}\mu$ for the pushforward measure. For a linear subspace $V \subset \mathbb{R}^d$, let $P_V$ denote the orthogonal projection onto $V$, and $V^\perp$ its orthogonal complement. We write

$$\mu^V := (P_V)_{\#}\mu, \qquad \mu^{V^\perp} := (P_{V^\perp})_{\#}\mu,$$

for the projected and residual laws. Finally, let $\mathcal{V}_{d,r} := \{ V \subset \mathbb{R}^d : \dim(V) = r \}$ denote the Grassmann manifold of $r$-dimensional linear subspaces.

**Quadratic Wasserstein distance and displacement geometry** For $\mu, \nu \in \mathcal{P}_2(\mathbb{R}^d)$, the squared quadratic Wasserstein distance is

$$W_2^2(\mu, \nu) := \inf_{\pi \in \Pi(\mu, \nu)} \int_{\mathbb{R}^d \times \mathbb{R}^d} \|x - y\|^2 \, \mathrm{d}\pi(x, y), \quad (1)$$

where $\Pi(\mu, \nu)$ is the set of couplings with marginals $(\mu, \nu)$. Throughout, we work with the quadratic cost and emphasize its geometric interpretation.

When $\mu$ and $\nu$ are absolutely continuous with respect to Lebesgue measure, Brenier's theorem (Brenier, 1991) ensures existence of an optimal transport map $T : \mathbb{R}^d \to \mathbb{R}^d$ such that $T_{\#}\mu = \nu$ and $(\mathrm{Id}, T)_{\#}\mu$ attains (1). In this case,

$$W_2^2(\mu, \nu) = \mathop{\mathbb{E}}_{X \sim \mu} \left[ \|T(X) - X\|^2 \right]. \quad (2)$$

Define the optimal displacement random vector $\Delta :=$ $T(X) - X$ for $X \sim \mu$ and its second-moment matrix

$$M(\mu, \nu) := \mathbb{E}\left[\Delta \Delta^\top\right] \in \mathbb{R}^{d \times d}. \tag{3}$$

Under (2), $\operatorname{tr} M(\mu, \nu) = W_2^2(\mu, \nu)$. The spectrum of $M(\mu, \nu)$ therefore quantifies how transport energy is distributed across orthogonal directions; this observation underlies the "spiked" transport structures used later. Since orthogonal projections are 1-Lipschitz, for any $V \in \mathcal{V}_{d,r}$,

$$W_2(\mu^V, \nu^V) \vee W_2(\mu^{V^\perp}, \nu^{V^\perp}) \leq W_2(\mu, \nu). \tag{4}$$

Here $a \vee b := \max\{a, b\}$. Moreover, the displacement matrix yields a convenient energy upper bound for projected discrepancies.

**Lemma 2.1** (Projected transport energy bound). *Assume $\mu, \nu$ are absolutely continuous and let $M(\mu, \nu)$ be as in* (3). *Then for any $V \in \mathcal{V}_{d,r}$, $W_2^2(\mu^V, \nu^V) \leq \operatorname{tr}\left(P_V M(\mu, \nu) P_V\right)$.*

**Projection pursuit and the WPP functional** High-dimensional plug-in estimators of $W_p$ suffer from the curse of dimensionality. Projection pursuit addresses this by searching for low-dimensional views on which the transport discrepancy concentrates. For $p \geq 1$ and $r \in \{1, \ldots, d\}$, define the *projection-pursuit Wasserstein functional*

$$\widetilde{W}_{p,r}(\mu, \nu) := \sup_{V \in \mathcal{V}_{d,r}} W_p(\mu^V, \nu^V). \tag{5}$$

The corresponding empirical estimator (Wasserstein projection pursuit, WPP) replaces $(\mu, \nu)$ with empirical laws, as in (Niles-Weed & Rigollet, 2022), and Proposition 2.2 recalls its finite-sample behavior.

**Proposition 2.2** (Typical finite-sample behavior of WPP). *Assume $p \in [1, 2]$ and that $\mu, \nu$ satisfy a transport inequality $T_p(\sigma^2)$. Let $\mu_n, \nu_n$ be empirical measures based on $n$ i.i.d. samples from $\mu$ and $\nu$, and define $\widehat{W}_{p,r} := \widetilde{W}_{p,r}(\mu_n, \nu_n)$. Then there exists $c_r > 0$ (depending on $p, r$ but not on $d, n$) such that*

$$\mathbb{E}\left[\left|\widehat{W}_{p,r} - \widetilde{W}_{p,r}(\mu, \nu)\right|\right] \leq c_r \, \sigma\left(r_{p,r}(n) + \sqrt{\tfrac{d \log n}{n}}\right), \tag{6}$$

*where $r_{p,r}(n) := c_p \sqrt{r} \times \begin{cases} n^{-1/(2p)}, & r < 2p, \\ n^{-1/(2p)}(\log n)^{1/p}, & r = 2p, \\ n^{-1/r}, & r > 2p. \end{cases}$*

**Sliced Wasserstein distance** The sliced Wasserstein distance averages one-dimensional transport costs over random directions. For $p \geq 1$ and $\alpha, \beta \in \mathcal{P}_p(\mathbb{R}^k)$, define

$$\operatorname{SW}_p^p(\alpha, \beta) := \int_{\mathbb{S}^{k-1}} W_p^p\left((\pi_\theta)_\# \alpha, (\pi_\theta)_\# \beta\right) d\sigma_k(\theta), \tag{7}$$

where $\pi_\theta(x) := \langle \theta, x \rangle$, and $\sigma_k$ is the Haar probability measure on $\mathbb{S}^{k-1}$. The following comparison is the key inequality we exploit on high-dimensional residual subspaces.

**Proposition 2.3** (Sliced lower bound for quadratic transport). *Let $k \geq 1$ and $\alpha, \beta \in \mathcal{P}_2(\mathbb{R}^k)$. Then $k \operatorname{SW}_2^2(\alpha, \beta) \leq W_2^2(\alpha, \beta)$.*

Such inequality becomes tight when the optimal displacement on $\mathbb{R}^k$ is directionally isotropic, so that the average one-dimensional squared displacement recovers $\|\Delta\|^2/k$ without loss. This "isotropy of the residual" will be the geometric condition under which our main lower certificate is near-tight.

# 3 Theoretical Analysis: Subspace–Slicing Certificates

This section develops the population and finite-sample theory for the conditioned subspace–slicing (CSS) certificate. Our goal is twofold. First, we establish that the two-term objective $V \mapsto W_2^2(\mu^V, \nu^V) + k \operatorname{SW}_2^2(\mu^{V^\perp}, \nu^{V^\perp})$ defines a *certified* lower bound on the true quadratic transport radius, and strictly improves the projection-only baseline whenever residual discrepancy is present. Second, we characterize when this lower certificate is *near-tight* and provide a rigorous theorem identifying (and stabilizing) the population-optimal subspace.

## 3.1 A Certified Two-Term Lower Bound

We begin with the strongest deterministic statement in the paper. It shows that the *sum* of a low-dimensional OT term and a scaled residual sliced term remains a valid lower bound on the full quadratic Wasserstein distance, both conditionally and unconditionally.

**Theorem 3.1** (lower certificate: conditional and unconditional). *Fix $r \in \{1, \ldots, d-1\}$ and $k = d - r$. For each $z$ and each $V \in \mathcal{V}_{d,r}$, define the CSS objective*

$$\mathcal{L}_z(V) := W_2^2\left(\mu_{1|z}^V, \mu_{0|z}^V\right) + k \operatorname{SW}_2^2\left(\mu_{1|z}^{V^\perp}, \mu_{0|z}^{V^\perp}\right),$$

*and let $\operatorname{LB}^\star(z) = \sup_{V \in \mathcal{V}_{d,r}} \mathcal{L}_z(V)$. Then:*

$$W_2^2\left(\mu_{1|z}, \mu_{0|z}\right) \geq \operatorname{LB}^\star(z) \qquad \textit{for all } z, \tag{8}$$

*and hence $W_2^2(\mu_1, \mu_0) \geq \mathbb{E}\left[\operatorname{LB}^\star(Z)\right]$.*

Inequality (8) certifies that *every* candidate subspace $V$ yields a valid two-term lower bound on the conditional transport radius. Taking the supremum produces the best certificate within the class. From the perspective of partial identification, Theorem 3.1 delivers a dimension-adaptive transport primitive that can be safely plugged into downstream PI optimization programs without violating validity.

It does not rely on a parametric model for conditional distributions. It is a deterministic inequality holding at the level of couplings. In particular, it does not assert that $\text{LB}^\star(z)$ is sharp; sharpness is addressed next by structural conditions.

A potential objection is that "adding a second nonnegative term" might look tautological. The key point is that one cannot add an arbitrary residual term without breaking certification. The residual term must itself be a valid lower bound for the residual Wasserstein distance on $V^\perp$; moreover, the scaling by $k$ is essential. Without the scaling, $\text{SW}_2^2$ typically decays as $d^{-1}$ in high dimension and becomes vacuous; Theorem 3.1 shows that the *properly scaled* sliced term provides a nontrivial, certified recovery of residual transport energy.

### 3.2 Population Bias Control via Residual Anisotropy

Theorem 3.1 guarantees validity but does not quantify tightness. We demonstrate under a spiked transport geometry, the only source of looseness is a single residual anisotropy deficit. For $k \geq 1$ and $\alpha, \beta \in \mathcal{P}_2(\mathbb{R}^k)$, define

$$\text{Def}_k(\alpha, \beta) := W_2^2(\alpha, \beta) - k\,\text{SW}_2^2(\alpha, \beta) \geq 0. \quad (9)$$

This deficit vanishes when residual transport is directionally isotropic, which will be illustrated further. The next lemma isolates the regime in which the transport problem genuinely decomposes into a low-dimensional "signal" and a residual, serving as the population-level backbone for our bias upper control.

**Lemma 3.2** (Additivity under orthogonal product structure). *Let $U \in \mathcal{V}_{d,r}$ and write $k = d-r$. Suppose $\mu_a = \mu_a^U \otimes \mu_a^{U^\perp}$ for $a \in \{0, 1\}$, where $\mu_a^U \in \mathcal{P}_2(U)$ and $\mu_a^{U^\perp} \in \mathcal{P}_2(U^\perp)$. Here $\otimes$ denotes the product measure induced by the orthogonal decomposition $\mathbb{R}^d = U \oplus U^\perp$. Then*

$$W_2^2(\mu_1, \mu_0) = W_2^2(\mu_1^U, \mu_0^U) + W_2^2(\mu_1^{U^\perp}, \mu_0^{U^\perp}).$$

*The same statement holds conditionally for $\{\mu_{1|z}, \mu_{0|z}\}$ whenever the product structure holds at each $z$.*

We can now upper bound the population certificate gap $\Delta(z) := W_2^2(\mu_{1|z}, \mu_{0|z}) - \text{LB}^\star(z) \geq 0$ by the residual anisotropy deficit. This is the population-level "bias upper control" promised in the introduction.

**Theorem 3.3** (Population bias upper control under spiked geometry). *Assume the conditional measures satisfy the product decomposition in Lemma 3.2 with signal subspace $U(z) \in \mathcal{V}_{d,r}$, and write $k(z) = d - r$. Then for each $z$, $0 \leq \Delta(z) := W_2^2(\mu_{1|z}, \mu_{0|z}) - \text{LB}^\star(z) \leq \text{Def}_{k(z)}\big(\mu_{1|z}^{U(z)^\perp}, \mu_{0|z}^{U(z)^\perp}\big)$. In particular, if the residual discrepancy is directionally isotropic so that $\text{Def}_{k(z)}(\cdot, \cdot) = 0$, then the CSS certificate is sharp at stratum $z$ i.e. $\Delta(z) = 0$.*

Theorem 3.3 says: once the dominant transport discrepancy is captured on a low-dimensional subspace, the only reason our lower bound is not exact is that the residual transport is anisotropic. In applications, a typical example is a learned embedding $X$ where treatment/control differences concentrate along a small number of "causal" directions, while the remaining coordinates behave like approximately whitened noise. In this regime, the residual transport is close to isotropic, hence $\text{Def}_k$ is small and the two-term certificate essentially recovers the full transport energy.

Theorem 3.3 does *not* assume that the residual discrepancy is zero. Rather, it shows that residual discrepancy can be *recovered* from one-dimensional projections up to an anisotropy deficit. This is precisely where projection-only methods fail: they discard the residual entirely, even in the benign isotropic regime where it is recoverable at almost no statistical cost.

We next give an interpretable sufficient condition that makes $\text{Def}_k$ small. The condition is expressed through the local spectral anisotropy of the residual Brenier map and leads to a dimension-adaptive, non-asymptotic bias control.

Let $T_\perp = \nabla\phi$ be the Brenier map from $\mu_{1|z}^{U(z)^\perp}$ to $\mu_{0|z}^{U(z)^\perp}$, and let $J(x) = \nabla T_\perp(x)$. For $X \sim \mu_{1|z}^{U(z)^\perp}$, define

$$\delta(J(x)) := 1 - \frac{\text{tr}(J(x))^2}{k(z)\,\text{tr}(J(x)^2)} \in [0, 1 - 1/k(z)],$$
$$\bar{\delta} := \mathbb{E}[\delta(J(X))].$$

The quantity $\delta(J(x))$ measures the local directional anisotropy of the residual Brenier map. It equals zero when $J(x)$ is proportional to the identity, and becomes larger when the residual transport stretches different directions unevenly. Thus $\bar{\delta}$ summarizes the average anisotropy of the residual transport over $\mu_{1|z}^{U(z)^\perp}$.

**Theorem 3.4** (Residual anisotropy implies small deficit). *Suppose the residual pair $\big(\mu_{1|z}^{U(z)^\perp}, \mu_{0|z}^{U(z)^\perp}\big)$ satisfies Assumption A.5. There exists a constant $C$ depending only on the regularity constants in the assumption such that*

$$\frac{\text{Def}_{k(z)}\big(\mu_{1|z}^{U(z)^\perp}, \mu_{0|z}^{U(z)^\perp}\big)}{W_2^2\big(\mu_{1|z}^{U(z)^\perp}, \mu_{0|z}^{U(z)^\perp}\big)} \leq C\sqrt{\bar{\delta}}.$$

*Consequently, the CSS certificate is near-tight whenever the residual Brenier map is close to directionally isotropic.*

### 3.3 Optimal Subspace and Recoverability

We now address the question: *which subspace should be used*, and does adding the residual sliced term distort subspace learning? Define $U^\star \in \arg\max_{V \in \mathcal{V}_{d,r}} \mathcal{L}_z(V)$. Under the extended spiked transport model, the transport energy concentrates on a low-dimensional signal space. The

next result shows that the CSS objective does *not* fundamentally change which subspace is optimal: under an eigen-gap and vanishing cross-displacement, the CSS optimizer remains close to the signal subspace.

*Assumption 3.5* (Extended spiked transport model (ESTM)). Fix $r \in \{1, \dots, d-1\}$ and let $k = d - r$. A pair $(\mu_0, \mu_1) \in \mathcal{P}_2(\mathbb{R}^d) \times \mathcal{P}_2(\mathbb{R}^d)$ satisfies *ESTM* if there exists a subspace $U \in \mathcal{V}_{d,r}$ and measures $\mu_{a,U} \in \mathcal{P}_2(U)$ and $\mu_{a,\perp} \in \mathcal{P}_2(U^\perp)$ such that $\mu_a = \mu_{a,U} \otimes \mu_{a,\perp}$ $(a \in \{0,1\})$, $\lambda_{\min}(M_{UU}) > \lambda_{\max}(M_{\perp\perp})$, where $M_{UU}$ and $M_{\perp\perp}$ are the displacement matrices associated with the transport pairs on $U$ and $U^\perp$, respectively.

**Theorem 3.6** (Stability of the CSS optimizer). *Assume the extended spiked transport model (Assumption 3.5) holds with signal subspace $U \in \mathcal{V}_{d,r}$. Assume moreover that the cross-block of the displacement matrix vanishes, $M_{U\perp} = 0$, when $M(\mu_0, \mu_1)$ is expressed in the orthogonal decomposition $U \oplus U^\perp$. Let $U^\star$ be any maximizer as above. Then*

$$\|P_{U^\star} - P_U\|_F^2 \le \frac{2k\,\lambda_{\max}(M_{\perp\perp})}{\lambda_{\min}(M_{UU}) - \lambda_{\max}(M_{\perp\perp})}. \quad (10)$$

*In particular, when the eigengap $\lambda_{\min}(M_{UU}) - \lambda_{\max}(M_{\perp\perp})$ is large, the CSS optimizer is uniquely determined up to small perturbations and aligns with the signal subspace.*

A common concern is that augmenting the objective with a residual term might "pull" the optimizer away from the true signal space. Theorem 3.6 shows this effect is controlled by an explicit eigengap: if the signal displacement energies dominate residual energies (the defining premise of spiked transport), then the optimizer is stable. Thus the residual slicing term improves the certificate *without* paying the price of unstable subspace learning in the regimes where dimension reduction is meaningful.

In all, the population theory suggests a practical two-stage strategy: (i) recover the dominant signal subspace using projection pursuit (WPP), and (ii) recover residual transport energy on the orthogonal complement using Monte Carlo sliced transport. The stability theorem justifies this decoupling: in spiked regimes, the subspace that optimizes the projection term is already near-optimal for the full CSS objective.

### 3.4 Finite-Sample Guarantees

We conclude the theory section with the nonasymptotic error bound for the plug-in CSS estimator in Algorithm 1. The bound separates the intrinsic-dimensional contribution (signal term) from the statistically stable residual contribution (sliced term).

**Theorem 3.7** (Finite-sample error of the CSS estimator). *Fix $z$ and suppose $\mu_{a|z}$ satisfy a transport inequality $T_{p'}(\sigma_a(z)^2)$ for some $p' \in [1,2]$. Let $\sigma(z) :=$*

---

**Algorithm 1** Conditioned subspace–slicing (CSS) certificate estimator

**Require:** Data $\{(X_i, A_i, Z_i)\}_{i=1}^N$, signal dimension $r$, number of slices $L$, and a procedure to form conditional empirical measures $\hat{\mu}_{a|z}$ (e.g. stratification or kernel weighting).

1: **for** each stratum (or query point) $z$ **do**
2:     Construct $\hat{\mu}_{1|z}$ and $\hat{\mu}_{0|z}$ from the samples.
3:     **(Signal subspace via WPP)** Compute $\hat{U}(z) \in \arg\max_{V \in \mathcal{V}_{d,r}} W_2^2\big((P_V)_\# \hat{\mu}_{1|z}, (P_V)_\# \hat{\mu}_{0|z}\big)$.
4:     **(Signal term)** Set $\widehat{S}(z) = W_2^2\big((P_{\hat{U}(z)})_\# \hat{\mu}_{1|z}, (P_{\hat{U}(z)})_\# \hat{\mu}_{0|z}\big)$.
5:     **(Residual term)** Let $\hat{\mu}_{a|z}^\perp = (P_{\hat{U}(z)^\perp})_\# \hat{\mu}_{a|z}$ and draw $\theta_\ell \overset{i.i.d.}{\sim} \mathrm{Unif}(\mathbb{S}^{k-1})$, $k = d - r$.
6:     Compute $\widehat{R}(z) = k \cdot \frac{1}{L}\sum_{\ell=1}^L W_2^2\big((\pi_{\theta_\ell})_\# \hat{\mu}_{1|z}^\perp, (\pi_{\theta_\ell})_\# \hat{\mu}_{0|z}^\perp\big)$.
7:     Output $\widehat{\mathrm{LB}}(z) = \widehat{S}(z) + \widehat{R}(z)$.
8: **end for**
9: Output the integrated certificate $\widehat{\overline{\mathrm{LB}}} = \frac{1}{N}\sum_{i=1}^N \widehat{\mathrm{LB}}(Z_i)$ (or an appropriate quadrature/average).

---

$\max\{\sigma_0(z), \sigma_1(z)\}$, $r(z) := \dim U(z)$, $k(z) := d - r(z)$, *and $n := \min\{n_0, n_1\}$. Let $\widehat{\mathrm{LB}}(z)$ be the estimator produced by Algorithm 1 with $L$ slices. Then there exists a universal constant $C > 0$ such that $\mathbb{E}\left[\left|\widehat{\mathrm{LB}}(z) - \mathrm{LB}(z)\right|\right] \le$*

$$C\,\sigma(z)^2 \left\{ r_{2,r(z)}(n) + \sqrt{\frac{d\log n}{n}} + k(z)\Big(\frac{1}{\sqrt{n}} + \frac{1}{\sqrt{L}}\Big) \right\},$$

*where $r_{2,r(z)}(n)$ is the intrinsic-dimensional rate in Proposition 2.2 (with $p = 2$).*

The WPP signal term contributes the intrinsic-dimensional rate $r_{2,r(z)}(n)$ (up to a lower-order ambient penalty), while the residual term scales as $k(z)(n^{-1/2} + L^{-1/2})$, reflecting that one-dimensional OT can be estimated at parametric rate and Monte Carlo slicing introduces an additional $L^{-1/2}$ term. Thus the CSS estimator recovers residual transport energy without reintroducing the curse of dimensionality.

Theorem 3.1 establishes validity (no false tightening). Theorem 3.3 and Theorem 3.4 explain when the certificate is near-tight (bias control). Theorem 3.6 guarantees that subspace learning is stable in spiked regimes (recoverability). Finally, Theorem 3.7 shows that the entire pipeline is statistically estimable in finite samples with intrinsic-dimensional scaling.

## 3.5 Application on Causal Partial Identification: Advantage of CSS Certificates

We finally clarify the causal origin of the transport problem studied as above and explains why the two-term certificate developed in Section 3 yields strictly more informative partial-identification bounds than projection-only or purely quadratic-moment surrogates in previous literature. The mapping from potential outcomes to couplings is standard and essentially measure-theoretic; the substantive difficulty is *high-dimensional* evaluation of the induced transport programs.

**Potential outcomes in Causality** We observe i.i.d. $(Y, X, A)$ with $A \in \{0, 1\}$ and potential outcomes $(Y(0), Y(1))$ taking values in $\mathbb{R}^{d_Y}$ (with $d_Y$ possibly large). Let $Z = h(X)$ be a (possibly vector-valued) covariate summary used for covariate assistance. Under standard conditions ensuring identification of *marginal* potential-outcome laws (e.g., random assignment, or conditional ignorability $(Y(0), Y(1)) \perp A \mid Z$), the conditional marginals $\nu_{a|z} := \text{Law}(Y(a) \mid Z = z)$ $(a \in \{0, 1\})$ are identified from the observed distribution via $\nu_{a|z} = \text{Law}(Y \mid A = a, Z = z)$. Partial identification arises because the *joint* conditional law of $(Y(0), Y(1))$ given $Z = z$ is not identified.

A particularly important class of partially identified causal functionals are *quadratic forms* of counterfactuals. They arise in dispersion/heterogeneity measures (e.g., the second moment of individual treatment effects), risk bounds, and variance-minimization formulations. These functionals induce OT (or multi-marginal OT) with quadratic costs.

**Proposition 3.8** (Quadratic dispersion as a quadratic OT functional). *Fix $z$ and assume $\nu_{0|z}, \nu_{1|z} \in \mathcal{P}_2(\mathbb{R}^{d_Y})$. Consider the quadratic dispersion functional*

$$\psi_2(z) := \mathbb{E}\left[ \|Y(1) - Y(0)\|^2 \mid Z = z \right],$$

*whose value is not identified because it depends on the unknown coupling of $(Y(0), Y(1)) \mid Z = z$. Then the sharp lower endpoint of $\mathcal{I}_{\|\cdot\|^2}(z)$ equals the quadratic Wasserstein cost:*

$$\inf_{\pi_z \in \Pi(\nu_{0|z}, \nu_{1|z})} \int \|y_1 - y_0\|^2 \, d\pi_z(y_0, y_1) = W_2^2(\nu_{1|z}, \nu_{0|z}). \tag{11}$$

*Equivalently, $\psi_2(z) \geq W_2^2(\nu_{1|z}, \nu_{0|z})$ for every feasible coupling.*

**Multi-world extensions.** When the estimand depends on $K \geq 2$ potential outcomes ("multiple worlds"), the coupling ambiguity becomes multi-marginal and the sharp identified set can be characterized by a multi-marginal optimal transport (MOT) program(Gao et al., 2024). The two-marginal quadratic radius (11) remains a canonical building block

---

**Algorithm 2** CSS-enhanced OT primitive for causal partial identification

**Require:** Data $\{(Y_i, X_i, A_i)\}_{i=1}^N$, covariate summary $Z = h(X)$, signal dimension $r$, number of slices $L$.
1: **for** each stratum (or query point) $z$ **do**
2:     Construct estimates $\hat{\nu}_{a|z} \approx \text{Law}(Y \mid A = a, Z = z)$ for $a \in \{0, 1\}$.
3:     Compute the CSS certificate $\widehat{\text{LB}}^Y(z)$ for $W_2^2(\nu_{1|z}, \nu_{0|z})$ using Algorithm 1 with input measures $(\hat{\nu}_{1|z}, \hat{\nu}_{0|z})$.
4:     Plug $\widehat{\text{LB}}^Y(z)$ into the OT-based PI objective/endpoints (e.g., quadratic-form bounds such as (11)).
5: **end for**
6: Aggregate over $z$ as required by the estimand (e.g., $\mathbb{E}[\cdot]$ over $Z$).

---

in such formulations, and is already the sharp endpoint for quadratic dispersion in the binary-treatment case.

Proposition 3.8 shows that the causal reduction itself is conceptually clean. The practical bottleneck is that evaluating the sharp OT program in (11) (and its conditional analogues) is statistically and computationally difficult in high dimension. This difficulty is also highlighted by work that studies estimation of OT/MOT objectives for quadratic costs: for instance, Gao et al. (2024) establish convergence rates for plug-in MOT estimators and identify regimes in which minimax-optimal behavior is attainable, underscoring that dimension is a fundamental obstacle.

**Relation to existing OT-based PI frameworks.** Our contribution is orthogonal to recent advances in OT-based partial identification. For instance, Ji et al. (2024) develop a model-agnostic inferential framework based on OT duality, emphasizing validity even when nuisance components are difficult to estimate. Gao et al. (2024) characterize sharp identified sets via multi-marginal OT and study plug-in estimation for quadratic costs. The present paper complements these works by isolating a distinct high-dimensional bottleneck: *dimension-adaptive evaluation of quadratic transport primitives*.

Finally, Algorithm 2 summarizes how CSS serves as a drop-in primitive in OT-based partial-identification pipelines. The procedure for estimating conditional marginals can follow any valid strategy, while CSS only replaces the high-dimensional quadratic transport evaluation step.

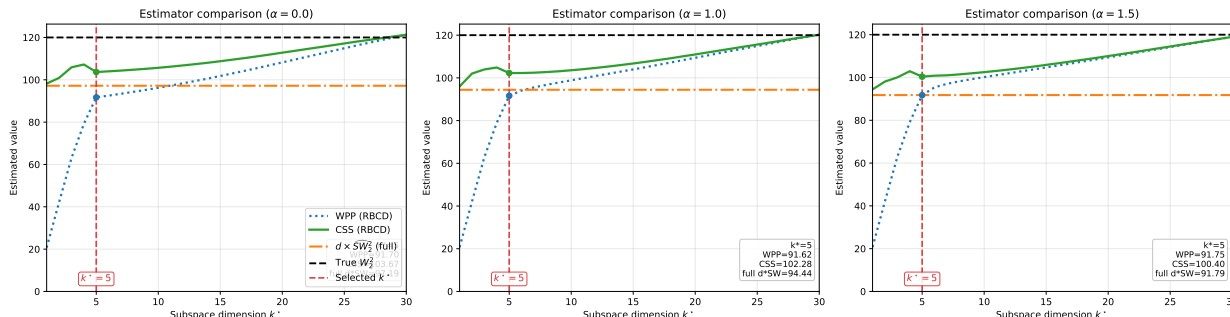

*Figure 1.* **Estimator comparison across subspace dimensions.** Synthetic Gaussian pushforward with $d = 30$, signal rank $r = 5$, and anisotropy $\alpha \in \{0, 1.0, 1.5\}$. Curves show WPP, CSS, the full SW baseline, and the analytic $W_2^2$ ground truth across candidate dimensions $k^\star$. The selected $k^\star$ is chosen by the WPP plateau rule. CSS stays closer to the true $W_2^2$ than projection-only WPP by adding a sliced residual correction on $U^\perp$. Optimization uses RBCD given by (Huang et al., 2021).

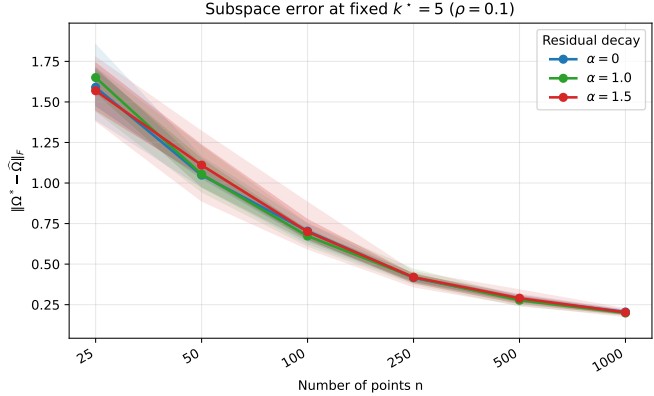

*Figure 2.* **Finite-sample subspace recovery at a fixed dimension.** We fix $k^\star = 5$ and report the projection error $\|\widehat{\Omega} - \Omega^\star\|_F$ as a function of the sample size $n$ for residual decay profiles $\alpha \in \{0, 1.0, 1.5\}$, with residual energy fraction $\rho = 0.1$. Curves show the median across trials, and shaded bands show quantile ranges. Optimization uses RBCD given by (Huang et al., 2021).

## 4 Experiments

In this section, we present numerical and real data experiments comparing the finite-sample performance of CSS with WPP and full sliced-Wasserstein baselines. WPP captures the dominant low-dimensional transport signal but discards residual transport energy, whereas full sliced-Wasserstein estimation averages one-dimensional transports over the ambient space. The code can be found in the link https://github.com/Map1e843/Causal-OT

### 4.1 Synthetic Data

We consider a synthetic setting where the ground truth Wasserstein distance and the optimal transport map are analytically tractable. Let the source measure be a standard Gaussian $\mu = \mathcal{N}(0, I_d)$ and the target measure be a pushforward $\nu = T_\# \mu$ defined under the linear map $T(x) = \Sigma^{1/2} x$, where $\Sigma \in \mathbb{R}^{d \times d}$ is a positive definite covariance matrix. Since $T$ is the gradient of the convex function $\psi(x) = \frac{1}{2} x^\top \Sigma^{1/2} x$, (Brenier, 1991) ensures that $T$ is the unique optimal transport map between $\mu$ and $\nu$. The squared Wasserstein distance is given by the Bures-Wasserstein met-

ric $\mathcal{W}_2^2(\mu, \nu) = \text{Tr}\big((\Sigma^{1/2} - I_d)^2\big)$. We construct the spectrum of the displacement matrix $M = (\Sigma^{1/2} - I_d)^2$ with dimension $d = 30$ and signal rank $r = 5$. Let signal eigenvalues be an equal constant without loss of generality, while the residual eigenvalues follow a power-law $j^{-\alpha}$ controlled by the anisotropy parameter $\alpha$. We fix the residual energy to be $10\%$ of the total energy which guarantee the eigengap condition in Assumption 3.5.

Figure 1 reports the estimator values produced by WPP, CSS, and the full sliced-Wasserstein baseline under the same synthetic Gaussian pushforward model. The WPP curve increases with $k^\star$ because larger projected subspaces capture more transport energy, but at the selected dimension ($k^\star = 5$) it still misses the residual discrepancy on $U^\perp$. CSS corrects this gap by adding the scaled sliced residual term, and therefore stays closer to the analytic $W_2^2$ ground truth across all three anisotropy levels. In contrast, the full $\widehat{d\text{SW}}_2^2$ baseline remains below the ground truth, reflecting the loss incurred by averaging over random directions in the ambient space. As $\alpha$ increases, the residual transport becomes more anisotropic and the CSS correction becomes slightly less tight, but it continues to improve over projection-only

WPP at the selected working dimension. We also use this experiment to illustrate our dimension selection rule. We inspect the WPP path and select the smallest dimension after the main marginal gain has saturated. In Figure 1, the WPP curve rises sharply up to $k^\star = 5$ and then enters a plateau, matching the planted signal rank. We therefore use $k^\star = 5$ as the working dimension and evaluate CSS at this dimension. This conservative choice avoids fitting high-dimensional projected OT when the additional gain is small, while CSS still recovers residual transport energy through the sliced correction on $U^\perp$.

Figure 2 fixes a conservative dimension $k^\star = 5$ and examines finite-sample recovery of the target subspace as $n$ increases. We report the projection error $\|\widehat{\Omega} - \Omega^\star\|_F$ under three residual decay profiles, $\alpha \in \{0, 1.0, 1.5\}$. Since the residual component accounts for only a small fraction of the total transport energy ($\rho = 0.1$), the leading five-dimensional signal subspace remains dominant across all three residual spectra. As a result, the absolute recovery curves are close, and we do not interpret Figure 2 as showing a strong ordering among the values of $\alpha$. Instead, the figure demonstrates finite-sample stability: the subspace error decreases consistently with $n$, from a relatively large error at small sample sizes to a much smaller error when $n = 1000$. The small separations between the curves indicate that residual spectral decay has a mild secondary effect, whereas the common convergence trend is driven primarily by the increasing sample size.

## 4.2 Real Data

We next evaluate CSS on the public RHC dataset. The treatment indicator records whether a patient received right-heart catheterization, and the outcome vector summarizes survival and hospitalization status over multiple follow-up horizons. This setting is useful for testing whether the proposed certificate remains informative beyond synthetic Gaussian models. Unlike the synthetic experiment, the real data experiment does not provide the population optimal transport map or the population quadratic Wasserstein distance. We therefore do not interpret this experiment as an absolute-error benchmark against the true $W_2^2$. Instead, we compare held-out lower certificates produced by WPP, CSS, and a full scaled sliced-Wasserstein baseline.

We construct a 27-dimensional outcome vector from three quantities at each horizon $h \in \{1, 2, 3, 5, 7, 10, 14, 21, 30\}$: survival beyond $h$ days, restricted survival time up to $h$ days, and hospital-free days up to $h$ days. All coordinates are normalized to a common scale. For each random split, we learn the WPP signal subspace on the training part and evaluate all certificates on the held-out part. To account for observed prognostic heterogeneity, we stratify patients by the SUPPORT survival-risk score and average the stratum-

level certificates using the held-out stratum masses. The same learned subspace is used for WPP and CSS; CSS differs only by adding the nonnegative sliced-Wasserstein correction on the orthogonal residual subspace.

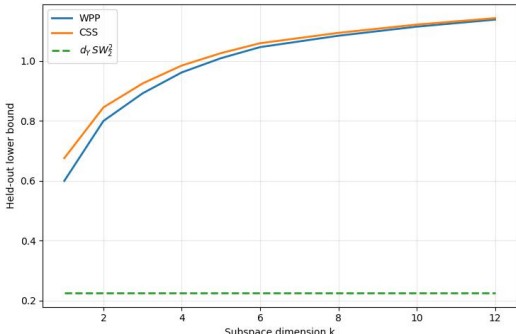

*Figure 3.* Real-data RHC comparison of WPP, CSS, and the full scaled sliced-Wasserstein baseline across subspace dimensions.

Figure 3 reports the held-out lower certificates as the subspace dimension varies. Across the tested dimensions, CSS lies above WPP. This is consistent with the theory: WPP captures the dominant low-dimensional transport signal but discards residual discrepancy, whereas CSS keeps the WPP signal term and recovers part of the residual transport energy through sliced one-dimensional projections. The gap between CSS and WPP is most visible at small and moderate dimensions, where more outcome variation remains in the orthogonal complement. Both WPP and CSS are also well above the full scaled sliced-Wasserstein baseline, suggesting that learning a task-adaptive signal subspace is important in this real-data problem. These results match the qualitative trend observed in the synthetic experiments. They show that CSS yields a more informative held-out transport certificate than projection-only WPP while retaining the conservative lower-bound interpretation.

## 5 Conclusion

In this paper, we propose the conditioned subspace–slicing (CSS) certificate to overcome the statistical fragility of OT-based causal partial identification in high dimensions. By integrating a low-dimensional OT signal with a scaled sliced-Wasserstein residual, our approach yields certified lower bounds that recover essential transport energy typically discarded by projection-only methods. We establish the tightness of this certificate under an extended spiked transport model and provide a plug-in estimator with nonasymptotic guarantees that explicitly characterize the intrinsic bias–variance trade-off. Empirical results confirm our theoretical insights: the CSS estimator effectively closes the population gap at the signal scale with minimal finite-sample overhead, offering a robust and dimension-adaptive primitive for high-dimensional causal inference.

## Acknowledgment

We are grateful to Zonghao Chen for helpful discussions. Zhiheng Zhang is supported by "the Fundamental Research Funds for the Central Universities" (Grant No. 2025110602) of Shanghai University of Finance and Economics, and Independent Research Project (Grant No. 2026110081) funded by the School of Statistics and Data Science. This work was supported by the Shanghai Engineering Research Center of Finance Intelligence (Grant No. 19DZ2254600).

## Impact Statement

This paper presents work whose goal is to advance the field of machine learning. There are many potential societal consequences of our work, none of which we feel must be specifically highlighted here.

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

# A   Proofs for Section 3

Throughout this appendix we work in Euclidean spaces equipped with the usual inner product and norm. For a linear subspace $V \subset \mathbb{R}^d$, let $P_V$ be the orthogonal projector onto $V$ and $V^\perp$ its orthogonal complement. For a Borel probability measure $\mu$ on $\mathbb{R}^d$ and a measurable map $T$, write $T_\#\mu$ for the pushforward of $\mu$ by $T$. For $V \subset \mathbb{R}^d$, we write $\mu^V := (P_V)_\#\mu$ and $\mu^{V^\perp} := (P_{V^\perp})_\#\mu$.

## A.1   Auxiliary Identities

**Lemma A.1** (Spherical second-moment identity). *Let $k \geq 1$ and $\sigma_k$ be the Haar probability measure on $\mathbb{S}^{k-1}$. Then*

$$\int_{\mathbb{S}^{k-1}} \theta\theta^\top \, \mathrm{d}\sigma_k(\theta) \;=\; \frac{1}{k}I_k, \qquad \text{and hence} \qquad \int_{\mathbb{S}^{k-1}} \langle\theta, v\rangle^2 \, \mathrm{d}\sigma_k(\theta) \;=\; \frac{\|v\|^2}{k} \text{ for all } v \in \mathbb{R}^k.$$

*Proof.* By rotational invariance of $\sigma_k$, the matrix $M := \int \theta\theta^\top \, \mathrm{d}\sigma_k(\theta)$ commutes with every orthogonal matrix, hence $M = cI_k$ for some $c \in \mathbb{R}$. Taking traces gives $1 = \int \mathrm{tr}(\theta\theta^\top) \, \mathrm{d}\sigma_k(\theta) = \mathrm{tr}(M) = ck$, hence $c = 1/k$. The second identity follows from $\langle\theta, v\rangle^2 = v^\top(\theta\theta^\top)v$. $\qquad\square$

**Lemma A.2** (Projector overlap and principal-angle identity). *Let $U, V \in \mathcal{V}_{d,r}$ and write $P_U, P_V$ for the orthogonal projectors. Then*

$$\|P_U - P_V\|_F^2 \;=\; 2r - 2\,\mathrm{tr}(P_U P_V) \;=\; 2\,\mathrm{tr}(P_U P_{V^\perp}) \;=\; 2\,\mathrm{tr}(P_V P_{U^\perp}).$$

*Proof.* Using $P_U^2 = P_U$, $P_V^2 = P_V$, and $\mathrm{tr}(AB) = \mathrm{tr}(BA)$,

$$\|P_U - P_V\|_F^2 = \mathrm{tr}\big((P_U - P_V)^2\big) = \mathrm{tr}(P_U) + \mathrm{tr}(P_V) - 2\mathrm{tr}(P_U P_V) = 2r - 2\mathrm{tr}(P_U P_V).$$

Since $P_{V^\perp} = I - P_V$ and $\mathrm{tr}(P_U) = r$, we have $r - \mathrm{tr}(P_U P_V) = \mathrm{tr}(P_U(I - P_V)) = \mathrm{tr}(P_U P_{V^\perp})$. The last equality follows similarly by symmetry. $\qquad\square$

## A.2   Sliced Wasserstein as a Scaled Lower Bound

**Proposition A.3** (Proof of Proposition 2.3). *Let $k \geq 1$ and $\alpha, \beta \in \mathcal{P}_2(\mathbb{R}^k)$. Define*

$$\mathrm{SW}_2^2(\alpha, \beta) := \int_{\mathbb{S}^{k-1}} W_2^2\big((\pi_\theta)_\#\alpha, (\pi_\theta)_\#\beta\big) \, \mathrm{d}\sigma_k(\theta), \qquad \pi_\theta(x) := \langle\theta, x\rangle.$$

*Then*

$$k\,\mathrm{SW}_2^2(\alpha, \beta) \;\leq\; W_2^2(\alpha, \beta).$$

*Proof.* Let $\pi^\star \in \Pi(\alpha, \beta)$ be an optimal coupling achieving $W_2^2(\alpha, \beta)$, and let $(X, Y) \sim \pi^\star$. Fix $\theta \in \mathbb{S}^{k-1}$. Then $(\pi_\theta(X), \pi_\theta(Y))$ is a coupling of $(\pi_\theta)_\#\alpha$ and $(\pi_\theta)_\#\beta$, so

$$W_2^2\big((\pi_\theta)_\#\alpha, (\pi_\theta)_\#\beta\big) \;\leq\; \mathbb{E}\big[\,|\langle\theta, X - Y\rangle|^2\,\big].$$

Integrating over $\theta$ and applying Lemma A.1 yields

$$\mathrm{SW}_2^2(\alpha, \beta) \leq \int_{\mathbb{S}^{k-1}} \mathbb{E}\big[\,|\langle\theta, X - Y\rangle|^2\,\big] \, \mathrm{d}\sigma_k(\theta) = \mathbb{E}\bigg[\int_{\mathbb{S}^{k-1}} |\langle\theta, X - Y\rangle|^2 \, \mathrm{d}\sigma_k(\theta)\bigg] = \frac{1}{k}\mathbb{E}\|X - Y\|^2.$$

Since $\mathbb{E}\|X - Y\|^2 = W_2^2(\alpha, \beta)$ under $\pi^\star$, the claim follows. $\qquad\square$

## A.3   Proof of Theorem 3.1

We first prove a slightly stronger pointwise statement: for each fixed $z$ and each fixed subspace $V$, the CSS objective $\mathcal{L}_z(V)$ is dominated by the full conditional quadratic transport radius.

**Lemma A.4** (Orthogonal decomposition lower bound). *Let $\mu, \nu \in \mathcal{P}_2(\mathbb{R}^d)$ and let $V \in \mathcal{V}_{d,r}$ with $k = d - r$. Then*

$$W_2^2(\mu, \nu) \geq W_2^2(\mu^V, \nu^V) + W_2^2(\mu^{V^\perp}, \nu^{V^\perp}).$$

*Consequently,*

$$W_2^2(\mu, \nu) \geq W_2^2(\mu^V, \nu^V) + k \, \mathrm{SW}_2^2(\mu^{V^\perp}, \nu^{V^\perp}).$$

*Proof.* Let $\pi \in \Pi(\mu, \nu)$ and let $(X, Y) \sim \pi$. Since $V \perp V^\perp$ and $P_V + P_{V^\perp} = I$, we have the Pythagorean identity

$$\|X - Y\|^2 = \|P_V(X - Y)\|^2 + \|P_{V^\perp}(X - Y)\|^2.$$

Taking expectation under $\pi$ gives

$$\mathbb{E}\|X - Y\|^2 = \mathbb{E}\|P_V(X - Y)\|^2 + \mathbb{E}\|P_{V^\perp}(X - Y)\|^2.$$

Let $\pi^V := (P_V, P_V)_\# \pi \in \Pi(\mu^V, \nu^V)$. Then $\mathbb{E}\|P_V(X - Y)\|^2 = \int \|u - v\|^2 \, \mathrm{d}\pi^V(u, v) \geq W_2^2(\mu^V, \nu^V)$. Likewise, letting $\pi^{V^\perp} := (P_{V^\perp}, P_{V^\perp})_\# \pi \in \Pi(\mu^{V^\perp}, \nu^{V^\perp})$ gives $\mathbb{E}\|P_{V^\perp}(X - Y)\|^2 \geq W_2^2(\mu^{V^\perp}, \nu^{V^\perp})$. Therefore, for every $\pi \in \Pi(\mu, \nu)$,

$$\int \|x - y\|^2 \, \mathrm{d}\pi(x, y) \geq W_2^2(\mu^V, \nu^V) + W_2^2(\mu^{V^\perp}, \nu^{V^\perp}).$$

Taking the infimum over $\pi$ yields the first inequality. The second follows by applying Proposition 2.3 on the residual space $V^\perp \simeq \mathbb{R}^k$. $\qquad\square$

*Proof of Theorem 3.1.* Fix $z$ and abbreviate $\mu := \mu_{1|z}$ and $\nu := \mu_{0|z}$. For any $V \in \mathcal{V}_{d,r}$, Lemma A.4 yields

$$W_2^2(\mu_{1|z}, \mu_{0|z}) \geq W_2^2(\mu_{1|z}^V, \mu_{0|z}^V) + k \, \mathrm{SW}_2^2(\mu_{1|z}^{V^\perp}, \mu_{0|z}^{V^\perp}) = \mathcal{L}_z(V).$$

Taking the supremum over $V \in \mathcal{V}_{d,r}$ gives (8).

**Integrated (in-$z$) form.** Since the inequality holds pointwise in $z$, we always have

$$\mathbb{E}\big[W_2^2(\mu_{1|Z}, \mu_{0|Z})\big] \geq \mathbb{E}\big[\mathrm{LB}^\star(Z)\big].$$

If one additionally assumes that the target quantity in the application is the integrated conditional radius $\mathbb{E}[W_2^2(\mu_{1|Z}, \mu_{0|Z})]$ (as in several covariate-assisted PI reductions), then this is precisely the desired unconditional statement.

**Remark on $W_2^2(\mu_1, \mu_0)$.** In general, $W_2^2(\mu_1, \mu_0) \geq \mathbb{E}[\mathrm{LB}^\star(Z)]$ need not hold for an arbitrary choice of $Z = h(X)$, because the optimal coupling between $\mu_1$ and $\mu_0$ may transport mass across different $z$-strata. The inequality $W_2^2(\mu_1, \mu_0) \geq \mathbb{E}[\mathrm{LB}^\star(Z)]$ does hold under the additional structural condition that there exists an optimal coupling $\pi^\star \in \Pi(\mu_1, \mu_0)$ such that $h(X) = h(Y)$ $\pi^\star$-a.s. (equivalently, the optimal plan is adapted to the partition induced by $h$); in that case, the cost decomposes stratum-wise and integration yields the claim. $\qquad\square$

## A.4  Proof of Lemma 3.2

*Proof of Lemma 3.2.* We make explicit that $\mu_{a,U} \otimes \mu_{a,\perp}$ denotes the product measure on $U \times U^\perp$, corresponding to independence between the $U$-component and the $U^\perp$-component.

Define the isometry $\mathsf{T} : U \times U^\perp \to \mathbb{R}^d$ by $\mathsf{T}(x, z) = x + z$. Orthogonality implies that for all $(x, z), (x', z') \in U \times U^\perp$,

$$\|\mathsf{T}(x, z) - \mathsf{T}(x', z')\|^2 = \|x - x'\|^2 + \|z - z'\|^2.$$

Let $\nu_a := \mu_{a,U} \otimes \mu_{a,\perp}$ on $U \times U^\perp$. Then $\mathsf{T}_\# \nu_a = \mu_a$ and $\mathsf{T}$ preserves the quadratic cost, hence

$$W_2^2(\mu_1, \mu_0) = W_2^2(\nu_1, \nu_0),$$

where the right-hand side is computed on $U \times U^\perp$ with cost $(x, z), (x', z') \mapsto \|x - x'\|^2 + \|z - z'\|^2$.

*Lower bound.* Let $\gamma \in \Pi(\nu_1, \nu_0)$ and write $((X_1, Z_1), (X_0, Z_0)) \sim \gamma$. Then

$$\mathbb{E}\left[\|X_1 - X_0\|^2 + \|Z_1 - Z_0\|^2\right] \geq W_2^2(\mu_{1,U}, \mu_{0,U}) + W_2^2(\mu_{1,\perp}, \mu_{0,\perp}),$$

since $(X_1, X_0)$ is a coupling of $(\mu_{1,U}, \mu_{0,U})$ and $(Z_1, Z_0)$ is a coupling of $(\mu_{1,\perp}, \mu_{0,\perp})$. Taking the infimum over $\gamma$ yields

$$W_2^2(\nu_1, \nu_0) \geq W_2^2(\mu_{1,U}, \mu_{0,U}) + W_2^2(\mu_{1,\perp}, \mu_{0,\perp}).$$

*Upper bound (attainment).* Let $\gamma_U^\star \in \Pi(\mu_{1,U}, \mu_{0,U})$ and $\gamma_\perp^\star \in \Pi(\mu_{1,\perp}, \mu_{0,\perp})$ be optimal couplings. Then $\gamma^\star := \gamma_U^\star \otimes \gamma_\perp^\star$ is a coupling in $\Pi(\nu_1, \nu_0)$ and achieves

$$\int \left(\|x - x'\|^2 + \|z - z'\|^2\right) \mathrm{d}\gamma^\star = W_2^2(\mu_{1,U}, \mu_{0,U}) + W_2^2(\mu_{1,\perp}, \mu_{0,\perp}).$$

Hence equality holds in the lower bound. Mapping back through $\mathsf{T}$ gives the claimed additivity on $\mathbb{R}^d$. The conditional statement follows identically by applying the argument at each fixed $z$. $\qquad\square$

## A.5 Proof of Theorem 3.3

*Proof of Theorem 3.3.* Fix $z$ and abbreviate $U := U(z)$ and $k := k(z)$. By definition,

$$\Delta(z) = W_2^2(\mu_{1|z}, \mu_{0|z}) - \mathrm{LB}^\star(z) \leq W_2^2(\mu_{1|z}, \mu_{0|z}) - \mathcal{L}_z(U),$$

since $\mathrm{LB}^\star(z) = \sup_V \mathcal{L}_z(V) \geq \mathcal{L}_z(U)$.

Under the product decomposition at $z$, Lemma 3.2 yields

$$W_2^2(\mu_{1|z}, \mu_{0|z}) = W_2^2\left(\mu_{1|z}^U, \mu_{0|z}^U\right) + W_2^2\left(\mu_{1|z}^{U^\perp}, \mu_{0|z}^{U^\perp}\right).$$

Therefore,

$$W_2^2(\mu_{1|z}, \mu_{0|z}) - \mathcal{L}_z(U) = W_2^2\left(\mu_{1|z}^{U^\perp}, \mu_{0|z}^{U^\perp}\right) - k\,\mathrm{SW}_2^2\left(\mu_{1|z}^{U^\perp}, \mu_{0|z}^{U^\perp}\right) = \mathsf{Def}_k\left(\mu_{1|z}^{U^\perp}, \mu_{0|z}^{U^\perp}\right),$$

which proves the upper bound on $\Delta(z)$. Nonnegativity $\Delta(z) \geq 0$ is immediate from Theorem 3.1. If the residual deficit vanishes, $\mathsf{Def}_k(\cdot, \cdot) = 0$, then $\Delta(z) = 0$ follows. $\qquad\square$

## A.6 Proof of Theorem 3.4

*Assumption A.5 (Regular residual transport).* Let $\alpha, \beta \in \mathcal{P}_2(\mathbb{R}^k)$ denote a residual transport pair. We assume:

1. Brenier regularity: The measures $\alpha$ and $\beta$ are absolutely continuous, and the quadratic optimal transport map from $\alpha$ to $\beta$ is $T = \nabla\varphi$.

2. Strong log-concavity of the source: The source $\alpha$ has density proportional to $\exp(-V)$, where $V$ is twice continuously differentiable and $\nabla^2 V(x) \succeq aI_k$ for some $a > 0$.

3. Uniform spectral bounds on the residual transport: The Brenier potential $\varphi$ is twice continuously differentiable $\mu$-a.e., and its Hessian $J(x) := \nabla^2\varphi(x)$ satisfies the uniform bounds $1 < m \leq L < \infty$ such that

$$mI_k \preceq J(x) \preceq LI_k \qquad \text{for } \alpha\text{-a.e. } x.$$

4. One-dimensional monotonicity: For $\sigma_k$-a.e. $\theta \in \mathbb{S}^{k-1}$, the regression map

$$u \mapsto \mathbb{E}[\langle \theta, T(X)\rangle \mid \langle \theta, X\rangle = u]$$

is nondecreasing.

5. Whitening: The residual source is centered and whitened: $\mathbb{E}[X] = 0$ and $\mathrm{Cov}(X) = I_k$ for $X \sim \alpha$.

In this subsection we work on a residual space $\mathbb{R}^k$. Let $\mu, \nu \in \mathcal{P}_2(\mathbb{R}^k)$ be absolutely continuous and let $T = \nabla\varphi$ be the Brenier map pushing $\mu$ to $\nu$. Let $X \sim \mu$ and define the displacement $\Delta := T(X) - X$ so that $W_2^2(\mu, \nu) = \mathbb{E}\|\Delta\|^2$. For $\theta \in \mathbb{S}^{k-1}$, define the scalar projections

$$U_\theta := \langle \theta, X \rangle, \qquad V_\theta := \langle \theta, T(X) \rangle = \langle \theta, X + \Delta \rangle.$$

**Lemma A.6** (Deficit as an average one-dimensional coupling gap). *With the above notation,*

$$\mathrm{Def}_k(\mu, \nu) = k \underset{\theta \sim \sigma_k}{\mathbb{E}} \left[ \mathbb{E}\,|V_\theta - U_\theta|^2 - W_2^2(\mu_\theta, \nu_\theta) \right],$$

*where $\mu_\theta := (\pi_\theta)_{\#}\mu$ and $\nu_\theta := (\pi_\theta)_{\#}\nu$.*

*Proof.* By definition and Lemma A.1,

$$W_2^2(\mu, \nu) = \mathbb{E}\|\Delta\|^2 = k\,\underset{\theta}{\mathbb{E}}\,\mathbb{E}\left[|\langle\theta, \Delta\rangle|^2\right] = k\,\underset{\theta}{\mathbb{E}}\,\mathbb{E}\,|V_\theta - U_\theta|^2.$$

On the other hand, $\mathrm{SW}_2^2(\mu, \nu) = \mathbb{E}_\theta\, W_2^2(\mu_\theta, \nu_\theta)$ by definition. Subtracting gives the identity. $\qquad\square$

**Lemma A.7** (Gap bound via monotone regression). *Assume that for $\sigma_k$-a.e. $\theta$ the regression function $m_\theta(u) := \mathbb{E}[V_\theta \mid U_\theta = u]$ is non-decreasing. Then for such $\theta$,*

$$\mathbb{E}\,|V_\theta - U_\theta|^2 - W_2^2(\mu_\theta, \nu_\theta) \leq 2\sqrt{\mathbb{E}\,|V_\theta - U_\theta|^2}\,\sqrt{\mathbb{E}\left[\mathrm{Var}(V_\theta \mid U_\theta)\right]}. \tag{12}$$

*Proof.* Fix $\theta$ and define $M_\theta := m_\theta(U_\theta)$, $\rho_\theta := \mathrm{Law}(M_\theta)$, and $d_\theta := W_2(\mu_\theta, \nu_\theta)$. By construction, the coupling $(U_\theta, M_\theta)$ is supported on the graph of the non-decreasing map $m_\theta$. In one dimension with quadratic cost, monotone rearrangement is optimal, so

$$W_2^2(\mu_\theta, \rho_\theta) = \mathbb{E}(M_\theta - U_\theta)^2 =: a_\theta^2.$$

Moreover, $(M_\theta, V_\theta)$ is a valid coupling of $\rho_\theta$ and $\nu_\theta$, hence

$$W_2^2(\rho_\theta, \nu_\theta) \leq \mathbb{E}(V_\theta - M_\theta)^2 = \mathbb{E}\left[\mathrm{Var}(V_\theta \mid U_\theta)\right] =: b_\theta^2.$$

The reverse triangle inequality gives

$$d_\theta \geq W_2(\mu_\theta, \rho_\theta) - W_2(\rho_\theta, \nu_\theta) \geq a_\theta - b_\theta.$$

Since also $d_\theta \geq 0$, we have

$$d_\theta \geq (a_\theta - b_\theta)_+, \qquad (x)_+ := \max\{x, 0\}.$$

Finally, by the conditional variance decomposition,

$$\mathbb{E}\,|V_\theta - U_\theta|^2 = \mathbb{E}(M_\theta - U_\theta)^2 + \mathbb{E}(V_\theta - M_\theta)^2 = a_\theta^2 + b_\theta^2.$$

Therefore,

$$\mathbb{E}\,|V_\theta - U_\theta|^2 - W_2^2(\mu_\theta, \nu_\theta) \leq a_\theta^2 + b_\theta^2 - (a_\theta - b_\theta)_+^2 \leq 2b_\theta\sqrt{a_\theta^2 + b_\theta^2}.$$

Substituting the definitions of $a_\theta$ and $b_\theta$ gives (12). $\qquad\square$

**Lemma A.8** (Conditional Poincaré on hyperplane fibers). *Assume $\mu$ has density $d\mu(x) = Z^{-1}e^{-V(x)}dx$ with $V \in C^2(\mathbb{R}^k)$ and $\nabla^2 V \succeq \alpha I_k$. Let $\theta \in \mathbb{S}^{k-1}$ and $U_\theta = \langle\theta, X\rangle$ for $X \sim \mu$. Then for every $C^1$ function $f : \mathbb{R}^k \to \mathbb{R}$ with $\mathbb{E}\,f(X)^2 < \infty$,*

$$\mathbb{E}\left[\mathrm{Var}(f(X) \mid U_\theta)\right] \leq \frac{1}{\alpha}\,\mathbb{E}\left[\|(I - \theta\theta^\top)\nabla f(X)\|^2\right].$$

*Proof.* Let $\mu(\cdot \mid U_\theta = u)$ denote a regular conditional version of $\mu$ given $U_\theta = u$. Since $\mu$ is $\alpha$-strongly log-concave on $\mathbb{R}^k$, its restriction to each affine hyperplane $\{x : \langle \theta, x \rangle = u\}$ remains $\alpha$-strongly log-concave in the tangential directions (the Hessian lower bound persists upon restriction). By the Brascamp–Lieb (or Bakry–Émery) inequality on the hyperplane, each conditional measure satisfies the Poincaré inequality

$$\operatorname{Var}_{\mu(\cdot \mid U_\theta = u)}(f) \leq \frac{1}{\alpha} \int \|(I - \theta\theta^\top)\nabla f(x)\|^2 \, \mu(dx \mid U_\theta = u).$$

Integrating both sides with respect to the law of $U_\theta$ yields the claim. $\qquad\square$

**Lemma A.9** (Spherical average of the tangential Jacobian action). *Let $A \in \mathbb{R}^{k \times k}$ be symmetric. Then*

$$\mathop{\mathbb{E}}_{\theta \sim \sigma_k} \left[ \|(I - \theta\theta^\top)A\theta\|^2 \right] = \frac{\operatorname{tr}(A^2)}{k+2} \, \delta(A), \qquad \delta(A) := 1 - \frac{\operatorname{tr}(A)^2}{k \operatorname{tr}(A^2)} \in \left[0, 1 - \frac{1}{k}\right].$$

*Proof.* Using symmetry of $A$,

$$\|(I - \theta\theta^\top)A\theta\|^2 = \|A\theta\|^2 - (\theta^\top A\theta)^2 = \theta^\top A^2 \theta - (\theta^\top A\theta)^2.$$

By Lemma A.1, $\mathbb{E}_\theta[\theta^\top A^2 \theta] = \operatorname{tr}(A^2)/k$. For the second term, the standard fourth-moment identity for $\theta \sim \sigma_k$ gives

$$\mathbb{E}_\theta[(\theta^\top A\theta)^2] = \frac{\operatorname{tr}(A)^2 + 2\operatorname{tr}(A^2)}{k(k+2)}.$$

Subtracting yields

$$\mathbb{E}_\theta \|(I - \theta\theta^\top)A\theta\|^2 = \frac{\operatorname{tr}(A^2)}{k} - \frac{\operatorname{tr}(A)^2 + 2\operatorname{tr}(A^2)}{k(k+2)} = \frac{k\operatorname{tr}(A^2) - \operatorname{tr}(A)^2}{k(k+2)} = \frac{\operatorname{tr}(A^2)}{k+2}\left(1 - \frac{\operatorname{tr}(A)^2}{k\operatorname{tr}(A^2)}\right),$$

as claimed. $\qquad\square$

**Lemma A.10** (Strong monotonicity implies a nontrivial transport scale). *Assume that $T = \nabla\varphi$ is $C^1$ and its Jacobian satisfies $\nabla T(x) \succeq mI_k$ for all $x \in \mathbb{R}^k$ with some $m > 1$. Then $T$ is $m$-strongly monotone:*

$$\langle T(x) - T(y), x - y \rangle \geq m\|x - y\|^2 \qquad \text{for all } x, y \in \mathbb{R}^k.$$

*If moreover $\operatorname{Cov}(X) = I_k$ for $X \sim \mu$, then*

$$W_2^2(\mu, \nu) = \mathbb{E}\|T(X) - X\|^2 \geq k(m-1)^2.$$

*Proof.* For $x, y \in \mathbb{R}^k$, by the fundamental theorem of calculus along the segment $y + t(x - y)$,

$$T(x) - T(y) = \int_0^1 \nabla T\big(y + t(x - y)\big)(x - y)\, dt.$$

Taking inner products with $(x - y)$ and using $\nabla T \succeq mI_k$ yields

$$\langle T(x) - T(y), x - y \rangle = \int_0^1 (x - y)^\top \nabla T(y + t(x - y))(x - y)\, dt \geq \int_0^1 m\|x - y\|^2 \, dt = m\|x - y\|^2.$$

For the lower bound on $W_2^2$, let $X, X' \overset{i.i.d.}{\sim} \mu$ and define $\Delta = T(X) - X$, $\Delta' = T(X') - X'$. Strong monotonicity gives

$$\langle T(X) - T(X'), X - X' \rangle \geq m\|X - X'\|^2.$$

Since $T(X) - T(X') = (X - X') + (\Delta - \Delta')$, we obtain

$$\langle \Delta - \Delta', X - X' \rangle \geq (m-1)\|X - X'\|^2.$$

Taking expectations and applying Cauchy–Schwarz yields

$$(m-1)\,\mathbb{E}\|X - X'\|^2 \leq \mathbb{E}\langle \Delta - \Delta', X - X' \rangle \leq \sqrt{\mathbb{E}\|\Delta - \Delta'\|^2} \sqrt{\mathbb{E}\|X - X'\|^2}.$$

Since $\operatorname{Cov}(X) = I_k$, we have $\mathbb{E}\|X - X'\|^2 = 2\operatorname{tr}(\operatorname{Cov}(X)) = 2k$. Thus $\mathbb{E}\|\Delta - \Delta'\|^2 \geq 2k(m-1)^2$. Finally, $\mathbb{E}\|\Delta - \Delta'\|^2 \leq 2\,\mathbb{E}\|\Delta\|^2$ (because $\mathbb{E}\|\Delta - \Delta'\|^2 = 2(\mathbb{E}\|\Delta\|^2 - \|\mathbb{E}\Delta\|^2) \leq 2\,\mathbb{E}\|\Delta\|^2$), hence $\mathbb{E}\|\Delta\|^2 \geq k(m-1)^2$. $\quad\square$

*Proof of Theorem 3.4.* Let $J(x) := \nabla T(x) = \nabla^2 \varphi(x)$ and define

$$r_{\text{eff}}(J(x)) := \frac{\text{tr}(J(x))^2}{\text{tr}(J(x)^2)}, \qquad \delta(J(x)) := 1 - \frac{r_{\text{eff}}(J(x))}{k}, \qquad \bar{\delta} := \mathbb{E}[\delta(J(X))].$$

By Lemma A.6 and Lemma A.7,

$$\begin{aligned}
\text{Def}_k(\mu, \nu) &= k \, \mathbb{E}_{\theta} \left[ \mathbb{E} \, |V_\theta - U_\theta|^2 - W_2^2(\mu_\theta, \nu_\theta) \right] \\
&\leq 2k \, \mathbb{E}_{\theta} \left[ \sqrt{\mathbb{E} \, |V_\theta - U_\theta|^2} \, \sqrt{\mathbb{E} \, \text{Var}(V_\theta \mid U_\theta)} \right] \\
&\leq 2k \, \sqrt{\mathbb{E}_{\theta} \mathbb{E} \, |V_\theta - U_\theta|^2} \, \sqrt{\mathbb{E}_{\theta} \mathbb{E} \, \text{Var}(V_\theta \mid U_\theta)},
\end{aligned}$$

where the last step is Cauchy–Schwarz in $\theta$.

We next bound the two factors. First, since $V_\theta - U_\theta = \langle \theta, \Delta \rangle$,

$$\mathbb{E}_{\theta} \mathbb{E} \, |V_\theta - U_\theta|^2 = \mathbb{E} \left[ \int_{\mathbb{S}^{k-1}} \langle \theta, \Delta \rangle^2 \, d\sigma_k(\theta) \right] = \mathbb{E} \left[ \frac{\|\Delta\|^2}{k} \right] = \frac{1}{k} W_2^2(\mu, \nu),$$

by Lemma A.1.

Second, apply Lemma A.8 with $f_\theta(x) := \langle \theta, T(x) \rangle$. Since $\nabla f_\theta(x) = J(x)\theta$ and $(I - \theta\theta^\top)\theta = 0$,

$$\mathbb{E} \, \text{Var}(V_\theta \mid U_\theta) \leq \frac{1}{\alpha} \, \mathbb{E} \, \|(I - \theta\theta^\top)J(X)\theta\|^2.$$

Averaging in $\theta$ and using Lemma A.9 gives

$$\mathbb{E}_{\theta} \mathbb{E} \, \text{Var}(V_\theta \mid U_\theta) \leq \frac{1}{\alpha} \, \mathbb{E} \left[ \mathbb{E}_{\theta} \|(I - \theta\theta^\top)J(X)\theta\|^2 \right] = \frac{1}{\alpha} \, \mathbb{E} \left[ \frac{\text{tr}(J(X)^2)}{k+2} \delta(J(X)) \right].$$

Using the spectral upper bound $J(x) \preceq L I_k$ yields $\text{tr}(J(X)^2) \leq k L^2$, hence

$$\mathbb{E}_{\theta} \mathbb{E} \, \text{Var}(V_\theta \mid U_\theta) \leq \frac{k L^2}{\alpha(k+2)} \bar{\delta}.$$

Combining the bounds,

$$\text{Def}_k(\mu, \nu) \leq 2k \sqrt{\frac{1}{k} W_2^2(\mu, \nu)} \sqrt{\frac{k L^2}{\alpha(k+2)} \bar{\delta}} = 2 \, W_2(\mu, \nu) \, \frac{kL}{\sqrt{\alpha(k+2)}} \, \sqrt{\bar{\delta}}.$$

Finally, Lemma A.10 implies $W_2(\mu, \nu) \geq (m-1)\sqrt{k}$ under the lower Jacobian bound $J \succeq m I_k$ and whitening $\text{Cov}(X) = I_k$. Therefore,

$$\frac{\text{Def}_k(\mu, \nu)}{W_2^2(\mu, \nu)} \leq \frac{2 \frac{kL}{\sqrt{\alpha(k+2)}} \sqrt{\bar{\delta}}}{(m-1)\sqrt{k}} = 2 \underbrace{\frac{L}{m-1} \sqrt{\frac{k}{\alpha(k+2)}}}_{=:C(\alpha, m, L, k)} \sqrt{\bar{\delta}},$$

which proves the theorem. $\qquad\square$

## A.7 Proof of Theorem 3.6

We work in the unconditional setting for clarity; the argument is identical conditionally at each fixed $z$. Let $M := M(\mu_0, \mu_1) = \mathbb{E}[\Delta\Delta^\top]$ be the displacement second moment matrix under an optimal coupling, and assume the ESTM product structure so that, in the orthogonal decomposition $U \oplus U^\perp$,

$$M = \begin{pmatrix} M_{UU} & 0 \\ 0 & M_{\perp\perp} \end{pmatrix}, \qquad \lambda_{\min}(M_{UU}) > \lambda_{\max}(M_{\perp\perp}).$$

**Lemma A.11** (Projected transport energy bound (Lemma 2.1))**.** *Assume $\mu_0, \mu_1$ are absolutely continuous and let $T$ be the Brenier map from $\mu_0$ to $\mu_1$. Let $\Delta := T(X) - X$ for $X \sim \mu_0$ and $M = \mathbb{E}[\Delta \Delta^\top]$. Then for any $V \in \mathcal{V}_{d,r}$,*

$$W_2^2(\mu_0^V, \mu_1^V) \leq \operatorname{tr}(P_V M P_V) = \operatorname{tr}(P_V M).$$

*Proof.* Under the Brenier coupling $(X, T(X))$, $(P_V X, P_V T(X))$ is a coupling between $\mu_0^V$ and $\mu_1^V$. Thus

$$W_2^2(\mu_0^V, \mu_1^V) \leq \mathbb{E}\, \|P_V(T(X) - X)\|^2 = \mathbb{E}[\Delta^\top P_V \Delta] = \operatorname{tr}(P_V\, \mathbb{E}[\Delta \Delta^\top]) = \operatorname{tr}(P_V M P_V).$$

$\square$

*Proof of Theorem 3.6.* Let $U^\star \in \arg\max_{V \in \mathcal{V}_{d,r}} \mathcal{L}(V)$, where

$$\mathcal{L}(V) := W_2^2(\mu_1^V, \mu_0^V) + k\,\mathrm{SW}_2^2(\mu_1^{V^\perp}, \mu_0^{V^\perp}), \qquad k = d - r.$$

By optimality, $\mathcal{L}(U^\star) \geq \mathcal{L}(U)$.

*Step 1: a curvature lower bound for the signal term.* Under the ESTM product structure, the optimal coupling decomposes along $U \oplus U^\perp$, hence

$$W_2^2(\mu_1^U, \mu_0^U) = \operatorname{tr}(M_{UU}).$$

For a general $V \in \mathcal{V}_{d,r}$, Lemma 2.1 and block-diagonality of $M$ give

$$W_2^2(\mu_1^V, \mu_0^V) \leq \operatorname{tr}(P_V M P_V) = \operatorname{tr}(P_V P_U M P_U P_V) + \operatorname{tr}(P_V P_{U^\perp} M_{\perp\perp} P_{U^\perp} P_V).$$

Let $B := P_U P_V P_U$ be the compression of $P_V$ to $U$. Then $0 \preceq B \preceq I_U$ and $\operatorname{tr}(B) = \operatorname{tr}(P_U P_V)$. Since $M_{UU} \succeq \lambda_{\min}(M_{UU}) I_U$,

$$\operatorname{tr}(M_{UU}) - \operatorname{tr}(B M_{UU}) = \operatorname{tr}((I_U - B) M_{UU}) \geq \lambda_{\min}(M_{UU}) \operatorname{tr}(I_U - B) = \lambda_{\min}(M_{UU}) \operatorname{tr}(P_U P_{V^\perp}).$$

Moreover, $M_{\perp\perp} \preceq \lambda_{\max}(M_{\perp\perp}) I_{U^\perp}$ implies

$$\operatorname{tr}(P_V P_{U^\perp} M_{\perp\perp} P_{U^\perp} P_V) \leq \lambda_{\max}(M_{\perp\perp}) \operatorname{tr}(P_V P_{U^\perp}).$$

Combining, and using $\operatorname{tr}(P_U P_{V^\perp}) = \operatorname{tr}(P_V P_{U^\perp})$, we obtain the lower bound

$$W_2^2(\mu_1^U, \mu_0^U) - W_2^2(\mu_1^V, \mu_0^V) \geq \big(\lambda_{\min}(M_{UU}) - \lambda_{\max}(M_{\perp\perp})\big) \operatorname{tr}(P_V P_{U^\perp}). \tag{13}$$

*Step 2: control the residual perturbation.* By Proposition 2.3 and Lemma A.4,

$$0 \leq k\,\mathrm{SW}_2^2(\mu_1^{V^\perp}, \mu_0^{V^\perp}) \leq W_2^2(\mu_1^{V^\perp}, \mu_0^{V^\perp}) \leq \operatorname{tr}(P_{V^\perp} M P_{V^\perp}) \leq \operatorname{tr}(M_{\perp\perp}) \leq k\,\lambda_{\max}(M_{\perp\perp}),$$

where we used $\dim(U^\perp) = k$ for the last inequality. Therefore,

$$k\,\mathrm{SW}_2^2(\mu_1^{U^{\star\perp}}, \mu_0^{U^{\star\perp}}) - k\,\mathrm{SW}_2^2(\mu_1^{U^\perp}, \mu_0^{U^\perp}) \leq k\,\lambda_{\max}(M_{\perp\perp}).$$

*Step 3: conclude by optimality.* Since $\mathcal{L}(U^\star) \geq \mathcal{L}(U)$,

$$W_2^2(\mu_1^U, \mu_0^U) - W_2^2(\mu_1^{U^\star}, \mu_0^{U^\star}) \leq k\,\mathrm{SW}_2^2(\mu_1^{U^{\star\perp}}, \mu_0^{U^{\star\perp}}) - k\,\mathrm{SW}_2^2(\mu_1^{U^\perp}, \mu_0^{U^\perp}) \leq k\,\lambda_{\max}(M_{\perp\perp}).$$

Applying (13) with $V = U^\star$ gives

$$\big(\lambda_{\min}(M_{UU}) - \lambda_{\max}(M_{\perp\perp})\big) \operatorname{tr}(P_{U^\star} P_{U^\perp}) \leq k\,\lambda_{\max}(M_{\perp\perp}).$$

Finally, Lemma A.2 implies $\operatorname{tr}(P_{U^\star} P_{U^\perp}) = \frac{1}{2}\|P_{U^\star} - P_U\|_F^2$, hence

$$\|P_{U^\star} - P_U\|_F^2 \leq \frac{2k\,\lambda_{\max}(M_{\perp\perp})}{\lambda_{\min}(M_{UU}) - \lambda_{\max}(M_{\perp\perp})},$$

which is (10). $\square$

## A.8   Proof of Theorem 3.7

We provide a proof of Theorem 3.7 in a form that is uniform over (possibly data-dependent) choices of subspaces, so it applies directly to Algorithm 1. Fix $z$ and abbreviate $\mu_a := \mu_{a|z}$. Let $n := \min\{n_0, n_1\}$ and let $\hat{\mu}_a$ be the empirical laws (or the conditional empirical laws produced by the conditioning scheme).

Write the population and empirical CSS quantities (at the chosen subspace) as

$$\mathrm{LB}(z) = S(z) + k\, R(z), \qquad \widehat{\mathrm{LB}}(z) = \widehat{S}(z) + k\, \widehat{R}_L(z),$$

where $\widehat{S}(z)$ is the WPP signal estimator (the maximized projected $W_2^2$ value), and $\widehat{R}_L(z)$ is the Monte Carlo sliced estimator in Algorithm 1.

**Lemma A.12** (One-dimensional plug-in stability). *Let $\lambda, \kappa \in \mathcal{P}_2(\mathbb{R})$ satisfy a transport inequality $T_{p'}(\sigma^2)$ for some $p' \in [1, 2]$, and let $\lambda_n, \kappa_n$ be empirical laws based on $n$ i.i.d. samples. Then there exists a constant $C_1 > 0$ (depending only on $p'$) such that*

$$\mathbb{E}\left|W_2^2(\lambda_n, \kappa_n) - W_2^2(\lambda, \kappa)\right| \leq C_1 \sigma^2 n^{-1/2}.$$

*Proof.* By the inequality $|a^2 - b^2| \leq (a + b)|a - b|$ and the reverse triangle inequality for $W_2$,

$$\left|W_2^2(\lambda_n, \kappa_n) - W_2^2(\lambda, \kappa)\right| \leq \left(W_2(\lambda_n, \kappa_n) + W_2(\lambda, \kappa)\right)\left|W_2(\lambda_n, \kappa_n) - W_2(\lambda, \kappa)\right|$$
$$\leq \left(W_2(\lambda_n, \kappa_n) + W_2(\lambda, \kappa)\right)\left(W_2(\lambda_n, \lambda) + W_2(\kappa_n, \kappa)\right).$$

Taking expectations and applying Cauchy–Schwarz yields

$$\mathbb{E}\left|W_2^2(\lambda_n, \kappa_n) - W_2^2(\lambda, \kappa)\right| \leq \sqrt{\mathbb{E}\left(W_2(\lambda_n, \kappa_n) + W_2(\lambda, \kappa)\right)^2}\sqrt{\mathbb{E}\left(W_2(\lambda_n, \lambda) + W_2(\kappa_n, \kappa)\right)^2}.$$

Under a $T_{p'}(\sigma^2)$ inequality with $p' \in [1, 2]$, standard empirical transport bounds in dimension one give $\mathbb{E}\, W_2(\lambda_n, \lambda) \lesssim \sigma n^{-1/2}$ and $\mathbb{E}\, W_2(\kappa_n, \kappa) \lesssim \sigma n^{-1/2}$, while $\mathbb{E}\, W_2(\lambda, \kappa) \lesssim \sigma$ and $\sup_n \mathbb{E}\, W_2(\lambda_n, \kappa_n) \lesssim \sigma$ (by triangle inequality and uniform integrability). Collecting these bounds gives the stated $n^{-1/2}$ rate for the squared quantity. $\square$

*Proof of Theorem 3.7.* We control the two components separately:

$$\mathbb{E}\left|\widehat{\mathrm{LB}}(z) - \mathrm{LB}(z)\right| \leq \mathbb{E}\left|\widehat{S}(z) - S(z)\right| + k\, \mathbb{E}\left|\widehat{R}_L(z) - R(z)\right|.$$

*Signal term.* By Proposition 2.2 (applied with $p = 2$ and dimension parameter $r(z)$), and using that orthogonal projections are 1-Lipschitz (hence preserve the $T_{p'}$ constants up to universal factors), there exists $C_2 > 0$ such that

$$\mathbb{E}\left|\widehat{S}(z) - S(z)\right| \leq C_2\, \sigma(z)^2 \left\{r_{2,r(z)}(n) + \sqrt{\frac{d \log n}{n}}\right\}.$$

*Residual sliced term.* Condition on the (possibly data-dependent) chosen subspace used in the residual step (in Algorithm 1 this is $\hat{U}(z)$). Conditional on this subspace, each $\theta_\ell$-projection produces a one-dimensional pair of measures, and by Lemma A.12,

$$\mathbb{E}\left[\left|W_2^2\left((\pi_{\theta_\ell})_\# \hat{\mu}_1^\perp, (\pi_{\theta_\ell})_\# \hat{\mu}_0^\perp\right) - W_2^2\left((\pi_{\theta_\ell})_\# \mu_1^\perp, (\pi_{\theta_\ell})_\# \mu_0^\perp\right)\right| \,\middle|\, \hat{U}(z), \theta_\ell\right] \leq C_1 \sigma(z)^2 n^{-1/2}.$$

Averaging over $\ell = 1, \ldots, L$ and then taking expectations yields the empirical component

$$\mathbb{E}\left|\frac{1}{L}\sum_{\ell=1}^{L} W_2^2\left((\pi_{\theta_\ell})_\# \hat{\mu}_1^\perp, (\pi_{\theta_\ell})_\# \hat{\mu}_0^\perp\right) - \frac{1}{L}\sum_{\ell=1}^{L} W_2^2\left((\pi_{\theta_\ell})_\# \mu_1^\perp, (\pi_{\theta_\ell})_\# \mu_0^\perp\right)\right| \leq C_1 \sigma(z)^2 n^{-1/2}.$$

For the Monte Carlo component, let $G(\theta) := W_2^2\left((\pi_\theta)_\# \mu_1^\perp, (\pi_\theta)_\# \mu_0^\perp\right)$. Then $R(z) = \mathbb{E}_\theta[G(\theta)]$ and

$$\mathbb{E}\left|\frac{1}{L}\sum_{\ell=1}^{L} G(\theta_\ell) - \mathbb{E}_\theta\, G(\theta)\right| \leq \sqrt{\frac{\mathrm{Var}(G(\theta))}{L}} \leq \sqrt{\frac{\mathbb{E}[G(\theta)^2]}{L}} \leq C_3\, \sigma(z)^2 L^{-1/2},$$

where the last inequality uses that $G(\theta) \le 2(\mathbb{E}\, U_\theta^2 + \mathbb{E}\, V_\theta^2)$ under an independent coupling and one-dimensional $T_{p'}$ control implies finite fourth moments bounded by constants times $\sigma(z)^4$. Combining empirical and Monte Carlo errors yields

$$\mathbb{E}\,|\widehat{R}_L(z) - R(z)| \;\le\; C_4\, \sigma(z)^2 \big(n^{-1/2} + L^{-1/2}\big).$$

Multiplying by $k(z)$ and combining with the signal bound gives (3.7)'s inequality. $\qquad\square$

## A.9 Proof of Proposition 3.8

*Proof of Proposition 3.8.* Fix $z$ and write $\nu_{a|z} := \mathrm{Law}(Y(a) \mid Z = z)$ for $a \in \{0, 1\}$. Any joint law of $(Y(0), Y(1)) \mid Z = z$ with these marginals corresponds to a coupling $\pi_z \in \Pi(\nu_{0|z}, \nu_{1|z})$. Therefore, the sharp lower endpoint of

$$\psi_2(z) = \mathbb{E}\left[\|Y(1) - Y(0)\|^2 \mid Z = z\right]$$

over all feasible joint laws equals the infimum of $\int \|y_1 - y_0\|^2 \,\mathrm{d}\pi_z(y_0, y_1)$ over $\pi_z \in \Pi(\nu_{0|z}, \nu_{1|z})$. By definition, this infimum is exactly $W_2^2(\nu_{1|z}, \nu_{0|z})$. $\qquad\square$

