# OpenReview forum: "Partial Identification under High-Dimensional Potential Outcomes and Confounders via  Optimal Transport"
_ICML.cc/2026/Conference — ICML 2026 regular_

### Official Review · Reviewer_QZyz · 2026-02-22

**Soundness:** 3
**Presentation:** 2
**Significance:** 3
**Originality:** 3
**Overall Recommendation:** 4
**Confidence:** 3

**Summary:**

This paper proposed a ``conditioned subspace-slicing'' method that provides squared quadratic Wasserstein distance between high-dimensional conditional distributions.
The main idea is to decomposes the transport into a low-dimensional signal subspace tern and a residual term via scaled sliced-Wasserstein distance on the orthogonal complement of the signal space.
The author proofed a lower bound, discuss the case when the inequality of the lower bound will hold.

**Compliance With Llm Reviewing Policy:**

Affirmed.

**Final Justification:**

The rebuttal have addressed my main concern and I would keep my recommendation.

**Key Questions For Authors:**

1. Is Proposition 2.2 derived by the authors? If not, please give the original references. I know some similar result like those in "Hütter, Jan-Christian, and Philippe Rigollet. "MINIMAX ESTIMATION OF SMOOTH OPTIMAL TRANSPORT MAPS." The Annals of Statistics 49.2 (2021): 1166-1194." So I would like to see the authors to provide some discussion on it.
2. How should $r$ be chosen in practice? Especially in the case of we know there is a trade-off in $r$. In real world application, how to guarantee the $r$ is optimally chosen?
3. I would suggest the authors to add some comparable method as baseline to illustrate the merits of the proposed method.

**Limitations:**

The major limitation is lack of real data analysis and comparisons with existing methods. Please see my comments above.

**Strengths And Weaknesses:**

Strength:

1. The OT transport problem is an important topic in statistics, and especially in high dimensional settings. The idea in this paper is novel and I saw its potential with applications in the subsequent analysis.
2. The core contribution is the lower bound, which combines the $W_2^2 $ distance of the signal part and a properly scaled residual sliced term. This result is important in understanding the behavior of OT in high dimensional space.
3. This is a well-written theoretical paper with informative experiment results that show the trade-off about the subspace dimension k. I would suggest the author consider put it in the main text ranter than the supplement since I think this is important.

Weakness:

1. My major criticism is the empirical results are somewhat limited. There is only an experiment based on the simulated data with Gaussian setting. I really like the idea in this paper but I would strongly suggest the author to include a real-world high dimensional data analysis to show the merits of the proposed method.
2. I may miss it but do the authors give examples or discussion on the conditional case? Since the authors mentioned conditional OT many times in the introduction, but later on seems focused on the unconditional case only.
3. The notations are confusing. E.g. line 65 the authors use $\mu^{\top}_{1\mid z}$ but later in line 148 such notations becomes $\mu^{V^{\top}}$.
4. The current version missed some important references in this field. E.g. Lin, Zhenhua, Dehan Kong, and Linbo Wang. "Causal inference on distribution functions." Journal of the Royal Statistical Society Series B: Statistical Methodology 85.2 (2023): 378-398, which studies causal inference through distributions.

---

> ### Author Rebuttal · Authors · 2026-03-29
>
> > **Q1: Is Proposition 2.2 original to the authors? If not, please cite the original source and clarify its relation to prior results, such as Hütter and Rigollet (2021).**
>
> **Answer:** Proposition 2.2 is not original to our paper. It is a restatement/adaptation of the finite-sample WPP analysis of Niles-Weed and Rigollet (2022) in our notation.
>
> Hütter and Rigollet (2021) study minimax estimation of the Brenier map under smoothness assumptions, whereas Proposition 2.2 concerns empirical estimation of the WPP functional, which exploits low-dimensional transport structure via projection pursuit. Thus the two are complementary: the former gives fast rates for full-map estimation, while WPP gives intrinsic-dimensional rates for transport discrepancy estimation under low-dimensional transport geometry.
>
> > **Q2: How should $r$ be chosen in practice? Especially in the case of we know there is a trade-off in r. In real world application， how to guarantee the $r$ is optimally chosen**
>
> **Answer:** We provide practical guidance for choosing $k^\ast$ in both the asymptotic and finite-sample regimes; empirically, the selected signal dimension is consistent with WPP.
>
> **1) Asymptotic case**
>
> In projected-OT, the projection dimension is usually treated as a structural tuning parameter. Prior SRW/PRW work suggests examining the projected-OT path across dimensions and selecting k via an elbow/plateau rule. Define
> $
> \widehat{\mathrm{WPP}}\_r := \sup\_{V \in \mathcal{V}\_{d,r}}W_2^2 \left(\hat{\mu}_1^V,\,\hat{\mu}\_0^V\right).
> $
>
> Under ESTM with an eigengap,
> $
> \Delta_r := \widehat{\mathrm{WPP}}_{r+1} - \widehat{\mathrm{WPP}}_r
> $
> drops sharply once $r$ exceeds the signal scale. In sufficiently large samples, this yields an observable elbow/plateau in the WPP curve, providing a natural heuristic for choosing $r$.
>
> [https://anonymous.4open.science/r/Causal-OT/css_n_10000.pdf]
>
> **2) Finite-sample case**
>
> A useful decomposition is
> $$
> \widehat{LB} - W_{2,\mathrm{true}}^2=\bigl(\widehat{LB}-LB^\star\bigr)+\bigl(LB^\star-W_{2,\mathrm{true}}^2\bigr),
> $$
> where the second term is always non-positive. The first term (finite-sample plug-in error) can be upward in small samples. Hence, in finite samples, `the key issue is the comparison between the upward plug-in fluctuation and the always negative gap`. This is why we advocate a more conservative choice of r when n is small: a smaller signal dimension typically reduces finite-sample fluctuation and leaves more slack, making raw plug-in validity easier to satisfy.
>
> [ https://anonymous.4open.science/r/Causal-OT/css_n_200.pdf.]
>
> >**Q3 & W1:  Some comparable method as baseline to illustrate the merits of the proposed method. And the emperical results are somewhat limited.**
>
> **Answer:** We have expanded the experiments for a more comprehensive evaluation (see the two links above). Beyond the dimension-selection study, we now include direct comparisons with multiple baselines (WPP and full sliced-Wasserstein) across different anisotropy settings. CSS consistently improves over WPP by recovering residual transport energy and stays much closer to the true $  W_2^2  $ when $  k^\ast  $ is properly chosen.
>
> We also add a real-data experiment on the public RHC dataset using a 27-dimensional outcome and honest train/test splits. CSS consistently dominates WPP across all tested subspace dimensions and both outperform the sliced-Wasserstein baseline.
>
> [https://anonymous.4open.science/r/Causal-OT/rhc_css.png]
>
> >**W2: Conditional case?**
>
> **R2:** Thank you for pointing this out. Our results are first stated conditionally: Theorem 3.1 defines $L_z(V)$ and shows $W_2^2(\mu\_{1|z},\mu\_{0|z})\ge LB^\star(z)$ for every $z$;` Lemma 3.2 and Theorem 3.3 likewise hold conditionally whenever the corresponding structure holds at each z`. The unconditional statements then follow by integrating over $Z$.
>
> We will revise the paper to introduce the conditional interpretation earlier with a brief roadmap and pointwise-in-$  z  $ example. For more delicate conditional regimes (rare strata or local non-overlap)we suggest a localized CSS version with covariate-assisted smoothing/borrowing across $z$; we will add this as a concrete future direction.
>
> > **W3: Notations are confusing.**
>
> **R3:** We have revised the manuscript to make the notation consistent throughout.
>
> > **W4: The current version missed some important references in this field.**
>
> **R4:** Thank you for highlighting this important reference. Lin, Kong, and Wang (2023) studies causal inference for distribution-valued outcomes in Wasserstein space, and is closely related to our work in connecting causal inference with distributional/OT-based analysis. While their focus is on estimation and inference for causal effects on distributions, our paper focuses on partial identification and high-dimensional OT-based causal bounds. We agree this paper is important for positioning our work in the broader literature, and we have added it to the revised manuscript.

---

> > ### Author Rebuttal · Reviewer_QZyz · 2026-04-02
> >
> > I will maintain my score.

---

> > > ### Author Response · Authors · 2026-04-04
> > >
> > > Thank you so much for this thoughtful and constructive suggestion! We have carefully and fully implemented your recommendation these days and will include it in our final version.

---

### Official Review · Reviewer_YJCF · 2026-03-04

**Soundness:** 3
**Presentation:** 2
**Significance:** 3
**Originality:** 3
**Overall Recommendation:** 4
**Confidence:** 4

**Summary:**

This work addresses the problem of conditional optimal transport distance estimation in high-dimensional settings. W2 distances between conditional outcome distributions play a key role in tightening partial identification bounds on functionals that relate to quadratic distances between counterfactuals, but naive estimators suffer from the curse of dimensionality. Existing approximation approaches tend to fall in two camps: (i) Wasserstein Projection Pursuit) (WPP), which lower bounds W2 by projecting the outcomes on a lower-dimensional subspace, (ii) Sliced Wasserstein (SW), which approximates W2 by an average of 1d projections. The authors propose a novel dimension-adaptive approximation to W2 between conditional outcome distributions (between treatment + control arm) that includes *both* a WPP signal term and scaled SW residual term. They prove this two-term sum is a (certified) lower bound, characterize its tightness via a residual anisotropy 'deficit' term, establish guarantees about subspace stability under a so-called "extended spiked transport model assumption", and derive finite-sample rates. Empirically, they show CSS generally improves on WPP based bounds and naive W2 estimators.

**Compliance With Llm Reviewing Policy:**

Affirmed.

**Final Justification:**

I thank the authors for addressing my concerns on the proof of Lemma B.6, and clarifying my understanding of several minor aspects.

My opinion of the paper has remained relatively unchanged since rebuttal. The authors present a novel and useful technique for lower bounding the $W2$ metric, with a view to using such bounds to improve estimation of partial identification bounds for for dispersion measures in cross-world counterfactual inference. The method is mathematically sound, and notably improves over the existing WPP and sliced techniques in most experiments.

The main limitations of the paper are somewhat inherent: since the techniques apply specifically to the $W2$ distance, one can only use the bound for a narrow range of cross-world quantities. (e.g., classic quantities like the median treatment effect and treatment harm rate are out of scope). The authors argued this could be extended to any quantities representable as $W2$ on feature embeddings, which is useful but remains somewhat specific.

Nonetheless, the contribution here extends beyond causal inference, to any setting where one desires a good finite-sample estimate of the $W2$ metric in high-dimensions.  I thus maintain my original score.

**Key Questions For Authors:**

(1) The authors state:

*"Consequently, the total error is U-shaped, providing a principled, data-driven choice of $k^*$ via minimizing the total-error curve".

Does one have access to this error curve in practice? My understanding is that one needs the true W2 distance to be able to compute this error curve.

(2) Can the authors comment on the expected performance (or mathematical relation) between  the proposed lower bound vs. using the slicing bound alone $kSW^2_2(\mu_{1|z}, \mu_{0|z})$? The proposed bound contains both a WPP and SW term, but the SW term is on the orthogonal subspace only so can't be directly compared to the naive SW bounding estimator in the same way. Obviously, in the isotropic case the naive SW approach is tight, and the anisotropic case motivates the present approach, however it isn't clear how bad this slicing estimator can be, and how the performance improvement may relate to the concentration of measure on subspace dimension.

(3) Can the authors provide some intuition on the ESTM assumption in Th 3.6 and when it is likely to fail/hold? It seems to mathematically formalize the notion of signal subspace dominance but it would be helpful to provide an example or two beyond the specific case in Sec 4.

(4) Can the authors check the error in Lemma B6 I pointed out above and address accordingly if needed.

**Limitations:**

In my view, there are a few limitations that could be discussed:

- Although the proposed lower bound is a direct improvement on WPP, as shown in experiments, finite-sample performance can be worse than using pure WPP, due to (what must be) estimation error. It would be useful to get some insight into why this is the case in Figure 1. Clearly the asymptotic convergence rate for this term is fast, but the scaling by $k$ may just inflate the constant sufficiently so that improvement over WPP is only expected after sufficiently many points (presumably $k$ may affect this too).

- While not specific to this method, the target functionals it applies to are all functionals of the for $E[||Y(1) - Y(0)||]$ and  their conditional analogues. While this includes important effects like the variance of the treatment effect and some dispersion measures, this is still a relatively niche selection of cross-world quantities. For example, the famous treatment harm rate (THR): $P(Y(1) - Y(0) < 0)$  or median treatment effect $Med(Y(1) - Y(0))$ is out of scope. This is more a criticism of the wider OT-PI framework but does limit the scope of impact and is worth mentioning.

- It is unclear how small the Deficit term is when the assumptions of Th 3.4 don't hold. (e.g., what about distributions which are heavier tailed, violating the log-concavity assumption?).

**Strengths And Weaknesses:**

**Soundness**

The paper appears technically sound except for one error I found in Lemma B6, which appears fixable. The main results provide certified lower bounds and so by construction hold in generality (thus hold under reasonable assumptions). The evaluation is fair, but it would be useful to see comparisons with the naive sliced bound aswell, and how performance varies with anisotropy and the extent to which the ESTM condition holds. The assumptions in Theorem 3.4 (which characterizes assumptions for closeness of the deficit term) are somewhat strong (e.g., log-concavity + uniformly conditioned jacobean), but do still cover a fairly wide range of distributions/situations, and these assumptions are commonly used in various results.


**Presentation**

The paper is generally well written, however I do think the presentation could be significantly improved. Below are some comments and recommendations:

- Section 1.1 (Lit review + positioning) is detailed but reads a bit too much like a taxonomy of related works, before the main ideas of the paper have been fleshed out properly. This may be a matter of taste, but it may be better to move some of this wider work to the later section in 3.5 on relationships to existing OT-based PI work, and merge the rest with the preliminaries. This way the paper could purely focus on the lines of work directly motivating the approach and preliminaries, saving the wider comparisons for after things are fully developed.
- Section 3 is far too large, despite the frequent use of subheadings. I would suggest splitting this into 2 (or even 3) sections. Something like (3.1-3.3) Bounds $\to$ (3.4) Estimation $\to$ (3.5) Applications etc, could work better for the flow of the paper.
- Section 3.5 again discusses relationships to existing OT-based PI work, having already laid out the background and positioning in Sec 1.1. I think this can work, but as mentioned above I would suggest restructuring.
- Theorem 3.4 takes nearly half a page to formulate, but is framed as providing intuition for when the crucial deficit term is small. I would strongly suggest moving the full theorem to the appendix and providing an informal and intuitive version in the main text.


**Significance**

The challenge of estimating Wasserstein distances in high-dimensions is clearly an important limitation of current OT-PI frameworks to address, and the proposed bound clearly improves on the existing approaches which use either sliced or subspace (WPP) approaches, making this a significant contribution in this area. The evaluation seems to show that the proposed bound tends to outperform the existing WPP approach in practice (e.g., in Fig 1 and Fig 2).

The significance is limited by (i) the fact that the finite-sample performance of the proposed estimator can be worse than basic WPP, as shown in Fig 2 and (ii) that the approach can only be used to improve PI bounds for specific cross-world functionals (i.e., those that relate to quadratic distances between counterfactuals). It is also not clear how small the deficit term is when the assumptions of Theorem 3.4 don't hold (since a lot of distributions are ruled out here).

**Originality**

The idea of decomposing the transport discrepancy  into a "low-dimensional signal" + "residual anisotropy" terms is (as far as I am aware) original and well-motivated. The main mathematical novelty is essentially finding a clever combination of two types of lower bound techniques for such transport costs into a modified bound that weakly improves on the WPP approaches, which I find very clean.

**Lemma B.6 error**

Just below eq.14  on pp.13 the authors use
$$W2 \geq a - b \implies W2^2 \geq (a-b)^2 $$
but this only holds in generality when $a-b > 0$. However I believe the reverse triangle inequality actually gives
$$W2 \geq |a - b|$$ in this situation, which fixes this issue.

---

> ### Author Rebuttal · Authors · 2026-03-29
>
> > **Q1: Access to curve in practice? Data-driven rule for signal dimension r**
>
> **Answer:**
> The U-shaped total-error curve in Fig. 1 is a synthetic illustration where true $W_2$ is known by construction, showing the bias–variance trade-off. In practice, we do not use unknown true $W_2$ to choose $k^\star$; we use data-driven rules based only on observable quantities. Full asymptotic and finite-sample discussion is in Reviewer 3, Q2.
>
> > **Q2: Proposed lower bound vs. slicing alone**
>
> **Answer:**
> We added a more comprehensive CSS-vs.-pure-slicing comparisonin https://anonymous.4open.science/r/Causal-OT/css_n_200.pdf, css_n_10000.pdf, and rhc_css.png.
> > **Q3: Intuition on ESTM in Thm. 3.6**
>
> **Answer:**
> `Intuitively, ESTM means treatment–control transport concentrates in a few coherent directions.` Discussion of why treatment–control differences may be anisotropic and low-dimensional is in our response to Reviewer 1, Q3.
>
> > **Q4: error in Lemma B.6**
>
> **Answer:**
> The reviewer is correct: squaring after the reverse triangle inequality is valid only for $a\_\theta\ge b\_\theta$. We will correct Lemma B.6 by defining
> $$
> a\_\theta^2:=\mathbb{E}(M\_\theta-U\_\theta)^2,\quad
> b\_\theta^2:=\mathbb{E}[\mathrm{Var}(V\_\theta\mid U\_\theta)],\quad
> d\_\theta:=W\_2(\mu\_\theta,\nu\_\theta),
> $$
> so that
> $$
> d\_\theta\ge (a\_\theta-b_\theta)\_+,\qquad (x)\_+:=\max(x,0).
> $$
> Since $\mathbb{E}|V_\theta-U_\theta|^2=a\_\theta^2+b\_\theta^2$,
> $$
> \mathbb{E}|V\_\theta-U\_\theta|^2-d\_\theta^2
> \le a\_\theta^2+b\_\theta^2-(a\_\theta-b\_\theta)\_+^2
> \le 2b\_\theta\sqrt{a\_\theta^2+b\_\theta^2}.
> $$
> In both cases: if $a_\theta>b_\theta$, it is the original argument; if $a_\theta\le b_\theta$, then
> $a\_\theta^2+b\_\theta^2-(a\_\theta-b\_\theta)\_+^2\le 2b\_\theta^2\le 2b\_\theta\sqrt{a\_\theta^2+b\_\theta^2}$.
> Hence
> $$
> \mathbb{E}|V_\theta-U_\theta|^2-W_2^2(\mu_\theta,\nu_\theta)
> \le 2\sqrt{\mathbb{E}|V_\theta-U_\theta|^2}\sqrt{\mathbb{E}[\mathrm{Var}(V_\theta\mid U_\theta)]}.
> $$
> Thus (12) and Theorem 3.4 remain unchanged; only the proof of Lemma B.6 is corrected.
>
> > **W1: presentation improvement**
>
> **R1:** We revised the manuscript point by point.
>
> > **W2: finite-sample performance can be worse than pure WPP**
>
> **R2:** We added a more comprehensive experiment in above links showing greater improvement.  Also the related data-driven rule of $k^\star$ can help solve it.
>
> > **W3: only improving PI bounds for quadratic cross-world functionals**
>
> **R3:**
> We agree non-smooth targets such as the treatment harm rate $P(Y(1)-Y(0)<0)$ and quantiles are important missing cases. `Our framework extends by keeping the OT/CSS certificate as the outer identification engine and replacing discontinuous targets by smooth dual surrogates`, as in Ji et al. (2024).
>
> For THR, bound $1\{u<0\}$ by smooth losses $\phi^-\_\tau(u)\le 1\{u<0\}\le \phi^+\_\tau(u)$, apply CSS, and obtain $L\_\tau\le P(Y(1)-Y(0)<0)\le U\_\tau$. The proof combines our lower-certificate argument with the dual-calibration step there; approximation error is controlled by $P(|Y(1)-Y(0)|\le\tau)$.
>
> With limited mass near zero, the surrogate gap is small, yielding valid and practically sharp harm-rate bounds without a new identification framework. The same applies to medians/quantiles by smoothing $F_\Delta(t)=P(Y(1)-Y(0)\le t)$ for each $t$ and inverting the bounds. Thus OT/CSS extends to broader functionals.
>
> ref: Model-Agnostic Covariate-Assisted Inference on Partially Identified Causal Effects.
> > **W4: Deficit size when Th. 3.4 assumptions fail**
>
> **R4:**
> A natural perturbation version: for any candidate signal subspace $U(z)$
> $$
> 0\le\Delta(z)\le A(z)+2\sqrt{k(z)}W_2\left(\mu_{1\mid z}^{U(z)^\perp},\mu_{0\mid z}^{U(z)^\perp}\right)\sqrt{\Gamma(z)},
> $$
> where
> $$
> A(z):=W_2^2(\mu_{1\mid z},\mu_{0\mid z})-W_2^2(\mu_{1\mid z}^{U(z)},\mu_{0\mid z}^{U(z)})-W_2^2(\mu_{1\mid z}^{U(z)^\perp},\mu_{0\mid z}^{U(z)^\perp})\ge0
> $$
> measures failure of exact signal–residual additivity, and
> $$
> \Gamma(z):=\mathbb{E}\_\theta[\operatorname{Var}(V\_{\theta,z}\mid U\_{\theta,z})]
> $$
> is the average one-dimensional conditional-variance defect of residual transport (Theorem 3.3 + Lemmas B.5–B.6). Thus mild violations of Theorem 3.4 do not cause breakdown; they yield explicit perturbation terms $A(z)$ and $\Gamma(z)$.
>
> Under strong log-concavity and Jacobian control, Theorem 3.4 gives
> $$
> \Gamma(z)\le \frac{k(z)L^2}{\alpha(k(z)+2)}\bar\delta(z),\qquad
> W_2\ge (m-1)\sqrt{k(z)},
> $$
> hence Theorem 3.4 is the sharpened special case
> $$
> \frac{\operatorname{Def}_{k(z)}}{W_2^2}\le C(\alpha,m,L,k(z))\sqrt{\bar\delta(z)}.
> $$
>
> If these assumptions fail, the clean $\sqrt{\bar\delta}$ formula weakens to the perturbation bound above, but remains. Validity is unchanged: for all $z$ and $V$,
> $$
> W_2^2(\mu_{1\mid z},\mu_{0\mid z})
> \ge W_2^2(\mu_{1\mid z}^V,\mu_{0\mid z}^V)+kSW_2^2(\mu_{1\mid z}^{V^\perp},\mu_{0\mid z}^{V^\perp}).
> $$
> CSS remains valid regardless of Theorem 3.4’s extra assumptions; only tightness is affected.

---

> > ### Author Rebuttal · Reviewer_YJCF · 2026-04-01
> >
> > Thank you for clarifying my questions and running the extra experiments. There are just two things remaining I would like clarified:
> >
> > 1. **New Experiment**: Thank you for providing this. I may have missed something obvious, but the plots show both WPP and your improved bound exceeding the raw $W2$ - why does this happen here?
> > 2. **Application to Other Cross-World Functionals:** My understanding is that the derived bounds are specific to the $W2$ distance, and hence to the specific choice of cost function $c(\cdot) = ||\cdot||^2$. Thus, it isn't clear to me how bounds on $W2$ via CSS can be used to bound $E[\phi(Y(1),Y(0))]$ for the arbitrary smooth $\phi$ used in either the upper or lower bound (unless $\phi = ||\cdot||^2)$.
> >
> > Many thanks.

---

> > > ### Author Response · Authors · 2026-04-04
> > >
> > > > **Comment 1: Both WPP and your improved bound exceeding the raw W2**
> > >
> > > **Response:**
> > > Thank you for pointing this out. The key distinction is between the population inequality and its finite-sample plug-in implementation. At the population level, our certificate is always a lower bound: for any subspace $V$,
> > > $$W_2^2(\mu^V, \nu^V) + kSW_2^2(\mu^{V^\perp}, \nu^{V^\perp}) \le W_2^2(\mu, \nu)$$
> > > What can exceed the plotted “raw $W_2$” is the empirical estimate, not the population target. In our figures, both the raw $W_2$ and the CSS/WPP curves are finite-sample estimates, and `finite-sample plug-in error can be upward in small samples`(see, e.g., Papp and Sherlock, Centered plug-in estimation of Wasserstein distances), especially in high-dimensional settings. Thus, this crossing should be understood as a finite-sample estimation effect rather than a violation of the population-level inequality.
> > >
> > > > **Comment 2: Application to other cross-world functionals.**
> > >
> > > **Response:**
> > > Before addressing the scope issue, we would like to emphasize that the quadratic case is already substantively meaningful, not merely a technical special case (cf. Gao et al., Bridging Multiple Worlds: Multi-marginal Optimal Transport for Causal Partial-identification Problem, AISTATS 2025). In particular, Theorem 3.1 provides a certified lower bound for the conditional quadratic Wasserstein distance, and Proposition 3.9 shows that for the quadratic dispersion
> > > $$
> > > \psi_2(z)=\mathbb E\left[|Y(1)-Y(0)|^2\mid Z=z\right],
> > > $$
> > > the sharp lower endpoint equals $W_2^2(\nu_{1\mid z},\nu_{0\mid z}).$
> > > So even within the quadratic class, the bound already applies to an important family of cross-world causal functionals.
> > >
> > > That said, the restriction is not to squared Euclidean cost in the raw outcome space itself. `What matters for our method is whether the part of the objective that depends on the coupling can be written, exactly or approximately, as a squared distance in some feature space.` We will discuss it in detail below.
> > >
> > > - **Kernel-induced cross-world functionals.** This extends naturally to kernel induced cross-world functionals. If for some feature map $\Phi: \mathcal{Y} \rightarrow \mathcal{H}\_k$
> > > $$
> > > \phi_k(y\_1,y\_0)=||\Phi(y\_1)-\Phi(y\_0)||_{\mathcal{H}\_k}^2= k(y\_1,y\_1)+k(y\_0,y\_0)- 2k(y\_1,y\_0),
> > > $$
> > > then the corresponding cross-world problem becomes a Wasserstein-2 problem after feature lifting:
> > > $$
> > > \inf\_{\pi\in\Pi(\nu\_1,\nu\_0)}\mathbb {E}\_\pi[\phi\_k(Y\_1,Y\_0)]=
> > > W\_{2,\mathcal {H}\_k}^2(\Phi (\nu\_1),\Phi (\nu\_0)).
> > > $$
> > > where $\Phi (\nu)$ denotes the pushforward of $\nu$ by $\Phi$.
> > > This is also consistent with the role of CSS in Section 3.5, where it is used in downstream causal PI problems.
> > >
> > > When the feature space is infinite-dimensional, if k admits a Mercer expansion $k(y,y')=\sum_{j\ge1}\lambda_j\psi_j(y)\psi_j(y')$, the canonical feature map is
> > > $$
> > > \Phi(y)=(\sqrt{\lambda_1}\psi_1(y),\sqrt{\lambda_2}\psi_2(y),\ldots)\in \ell_2.
> > > $$
> > > Truncating to the first $m$ components yields $\Phi_m(y)\in\mathbb R^m$ and cost
> > > $$
> > > c\_{k,m}(y_1,y_0)=||\Phi\_m(y\_1)-\Phi\_m(y\_0)||\_2^2=\sum_{j=1}^m \lambda\_j(\psi\_j(y\_1)-\psi\_j(y\_0))^2,
> > > $$
> > > After truncation, the problem reduces to the same finite-dimensional quadratic OT form that CSS certifies.
> > >
> > > - **General smooth $\phi$: approximation by polynomials**
> > > By the Stone-Weierstrass theorem, for any compact set $K \subset \mathbb{R}^d$, any smooth function $\phi\in C(K)$ can be uniformly approximated by polynomials. Thus one may write,
> > > $$\phi(y_1, y_0)\approx p_m(y_1, y_0)$$
> > > where $p_m$ is a polynomial in $(y_1,y_0)$. If $y(0) \in \mathbb{R}^{d_0}$ and $y(1) \in \mathbb{R}^{d_1}$, any polynomial can be written as
> > > $$p_m(y(0), y(1)) = \sum\_{\alpha, \beta} c\_{\alpha, \beta} \ y(0)^\alpha y(1)^\beta$$
> > > where $\alpha\in\mathbb N^{d_0}$ and $\beta\in\mathbb N^{d_1}$ are multi-indices, with
> > > $$
> > > y(0)^\alpha=\prod\_{j=1}^{d_0} y\_j(0)^{\alpha_j},\qquad y(1)^\beta=\prod_{k=1}^{d_1} y_k(1)^{\beta_k}.
> > > $$
> > > Collecting the monomials of $y(0)$ and $y(1)$, there exist finite-dimensional feature maps $\phi_{0,m}$ and $\phi_{1,m}$ into a common inner-product space such that
> > > $$p_m(y(0),y(1)) = \langle\phi_{0,m}(y(0)),\phi_{1,m}(y(1))\rangle$$
> > > Equivalently, writing $\widetilde\phi_{0,m}(y(0))=-\phi_{0,m}(y(0))$,
> > > $$
> > > p_m(y(0),y(1))=\frac{1}{2}||\phi\_{1,m}(y(1))-\widetilde\phi\_{0,m}(y(0))||^2
> > > -\frac{1}{2}||\phi\_{1,m}(y(1))||^2
> > > -\frac{1}{2}||\widetilde\phi\_{0,m}(y(0))||^2 .
> > > $$
> > > Therefore, under fixed marginals, minimizing $\mathbb E_\pi[p_m(Y(1),Y(0))]$ over couplings reduces to a problem whose coupling-dependent part is
> > > $$\frac{1}{2}\inf_{\pi\in\Pi(\nu_1,\nu_0)} \mathbb{E}\_\pi\left[
> > > ||\phi\_{1,m}(Y(1))-\widetilde\phi\_{0,m}(Y(0))||^2
> > > \right].$$
> > > The remaining terms depend only on the marginals. Hence, the minimization problem for the smooth functional finally reduces to $W_2$ framework studied in our paper. We present this as a natural extension route, rather than as a theorem proved here.

---

### Official Review · Reviewer_8zeT · 2026-03-13

**Soundness:** 3
**Presentation:** 2
**Significance:** 3
**Originality:** 3
**Overall Recommendation:** 4
**Confidence:** 2

**Summary:**

This paper studies causal partial identification when the relevant optimal transport quantities become hard to estimate in high dimensions. To address this, the paper proposes a conditioned subspace-slicing approach that decomposes transport into a low-dimensional signal component, estimated through Wasserstein projection, and a residual component on the orthogonal complement. The paper proves that the CSS objective is always a valid lower certificate, and that it is near-tight when characterized using an extended spiked transport model and a residual anisotropy deficit. Empirically, the paper uses synthetic Gaussian experiments to illustrate the predicted bias-variance tradeoff and shows that CSS recovers transport energy missed by projection-only baselines.

**Compliance With Llm Reviewing Policy:**

Affirmed.

**Final Justification:**

The authors have answered my question adequately and indicated improvement in the presentation. My concerns are addressed.

**Key Questions For Authors:**

1. Could you please explain more about the data-driven rule for the signal dimension $r$? The paper discusses a bias–variance tradeoff and shows U-shaped error curves in the synthetic experiments, but I don't fully understand the concrete selection rule here.
2. How robust is the proposed method when the residual component is not close to isotropic?
3. Could you please elaborate a bit more about the motivation for how the treatment–control distributional differences should be anisotropic and concentrated on a low-dimensional subspace?

**Limitations:**

I think the paper can discuss about how does the main assumption will hold in practice, and discuss more about how would the practical downstream tasks can be benefit from the proposed method.

**Strengths And Weaknesses:**

**Strength**

- The proposed method is well-motivated. The paper demonstrate the CSS fixes a limitation of the projection-only approach by recovering residual transport energy.
- The paper provides a clear intuition that ties the gap of existing methods to residual anisotropy, which makes the theory interpretable.
- The paper provides a novel contribution. Although there are existing works on OT-based partial ID for causal inference, this paper introduces novelty in the CSS decomposition and its analysis.

**Weakness**

- The introduction is quite dense with many theoretical terms and notions that are introduced later.
- It is not straightforward to assess how the main assumption will hold in practice.
- The evaluation was mainly with synthetic Gaussian data, which is a bit narrow.

---

> ### Author Rebuttal · Authors · 2026-03-29
>
> ### `A Kind General Response to All Reviewers and Dear AC`
>
> We thank the reviewers for their positive feedback on the `novelty, technical contribution, and analysis of our paper` and constructive suggestions. In the revised manuscript, we have addressed the main concerns as follows.
> 1. **Practical Clarifications and Assumptions:**
>  - Data-driven choice of signal dimension, robustness to anisotropic residuals, and intuition for low-dimensional transport structure (`8zeT-Q1–3, YJCF-Q1, QZyz-Q2`).
> 2. **Theoretical and Presentation Improvements:**
>  - Correcting Lemma B.6, clarifying ESTM/conditional case, improving notation, presentation, and references (`YJCF-Q3–4, QZyz-Q1, W2–4`).
> 3. **Expanded Empirical Validation:**
> - More baselines, anisotropy studies, and real-data experiments.(`YJCF-W2–4, QZyz-W1`).
>
>
>
> > ***Q1: the data-driven rule for the signal dimension $r$?***
>
> **Answer:** We answer this question from both asymptotic and finite-sample perspectives. Due to space limits, we refer to Reviewer 3 Q2 for details.
>
> >***Q2: How robust is the proposed method when the residual component is not close to
> isotropic?***
>
>
> **Answer:** We'll respond from two aspects: validity and tightness.
>
> 1. **Validity:** CSS remains valid regardless of residual isotropy. By Theorem 3.1, for all $z$ we have $W_2^2(\mu_{1|z}, \mu_{0|z}) \ge LB^\star(z).$
> Moreover, since the sliced correction is nonnegative,  $L_z(V) \ge W_2^2(\mu_{1|z}^V, \mu_{0|z}^V),$ Thus, for any candidate subspace, CSS is never looser than projection-only WPP.
>
> 2. **Tightness:**  Residual isotropy affects tightness. Under the orthogonal product decomposition in Lemma 3.2, Theorem 3.3 gives
> $$0 \le \Delta(z) := W_2^2(\mu_1|_z,\mu_0|_z)-LB^\star(z)\le \operatorname{Def}(k(z)) \big(\mu_1|_z^{U(z)^\perp},\mu_0|_z^{U(z)^\perp})$$
> so near-isotropy implies a small certificate gap (the upper bound is more informative). For the non-isotropic Gaussian case, we have
> $$W_2^2(\mu, \nu) - LB^\star \ge \frac{k\ tr(\Sigma\_\perp^2) - tr(\Sigma\_\perp)^2}{4(k+2)}$$
>
> Thus, `lack of residual isotropy does not invalidate CSS, but can worsen tightness`; both this gap and the Def term vanish in the isotropic case.
>
> >**Q3: the motivation for how the treatment–control distributional differences should be
> anisotropic and concentrated on a low-dimensional subspace?***
>
> **Answer:** A standard view in the WPP literature is that differences between high-dimensional distributions often concentrate in a low-dimensional subspace, as formalized by Niles-Weed and Rigollet’s spiked transport model. Projection and subspace-robust Wasserstein methods build on this idea. Our work follows it by capturing the dominant signal in a subspace and adding a residual sliced term rather than discarding the complement.
>
> This principle applies naturally in causal NLP (where high-dimensional text requires low-dimensional treatment- and outcome-relevant representations) and multi-omic/biomarker studies (where Essential Regression uses latent factors to link molecular measurements to outcomes through a few key biological pathways). These examples show that `“high-dimensional observations + lower-dimensional intervention-relevant structure” is standard and practically meaningful.` Our ESTM/CSS framework directly implements this principle in transport geometry.
>
>
> >**W1: The introduction is quite dense with many theoretical terms and notions that are introduced later.**
>
> **R1:** We revised the introduction by moving technical notions (CSS certificate, spiked transport model, signal–residual decomposition) later. We now first explain the problem setting and core intuition, and add a clearer roadmap separating motivation, the CSS idea, and the main results before formal definitions.
>
> >**W2: It is not straightforward to assess how the main assumption will hold in practice.**
>
> **R2:** We clarified how to assess the assumptions in practice, e.g., by checking fast eigenvalue decay in sample covariances and stability of the leading projection subspace across resamples/splits. We also emphasize the assumption hierarchy: CSS validity requires only weak moment conditions, while stronger assumptions (approximate product structure, near-isotropic residuals, eigengap) mainly improve tightness, subspace recovery, and finite-sample guarantees. They can be relaxed at the cost of looser conclusions.
>
> >**W3 The evaluation was mainly with synthetic Gaussian data, which is a bit narrow.**
>
> **R3:** The Gaussian example is used only because it provides a closed-form expression for the certificate gap. CSS validity is distribution-free at the population level, following from the deterministic inequality $kSW\_2^2 \le W\_2^2$ independent of Gaussianity.  More generally, under the low-rank signal + residual decomposition, the gap bound extends to the sub-Gaussian case in the same form as above. Thus, near-isotropic sub-Gaussian residuals yield a small gap, while anisotropic residuals lead to a larger but still controlled looseness.

---

> > ### Author Rebuttal · Reviewer_8zeT · 2026-04-02
> >
> > Thanks to the authors for the clarifications. My questions are well addressed in the rebuttal. My evaluation remain unchanged.

---

> > > ### Author Response · Authors · 2026-04-04
> > >
> > > Thank you so much for this thoughtful, constructive suggestion and your praise in the final justification! We have carefully and fully implemented your recommendation these days, and we have included it in the final version.

---

### Decision · Program_Chairs · 2026-04-30

**Decision:**

Accept (regular)

**Comment:**

The work proposes a novel estimator suitable for OT in high dimensions. The key idea is to decompose the transport problem into
a low-dimensional signal subspace and a high-dimensional residual subspace. The estimation in high-dimensional residual space is outsourced to methods like sliced Wasserstein. The analysis around the idea is self-complete and further the technical contribution of the work. Though the application proposed is causal inference, the improved methodology can have bearings on other problems. It will be interesting to see if such ideas may help OT based strategies to compete with SOTA in generative models.

The reviewers suggestion have helped improve the presentation and few gaps that existed in the earlier stage.

Given the positive feedback from reviewers, the novel idea and its sound analysis, and potential applications to other problems like generative models, I recommend accepting this paper.